# Putting the theory into 'burstlet theory' with a biophysical model of burstlets and bursts in the respiratory preBötzinger complex

**Ryan S Phillips\*, Jonathan E Rubin\***

Department of Mathematics and Center for the Neural Basis of Cognition, University of Pittsburgh, Pittsburgh, United States

**Abstract** Inspiratory breathing rhythms arise from synchronized neuronal activity in a bilaterally distributed brainstem structure known as the preBötzinger complex (preBötC). In in vitro slice preparations containing the preBötC, extracellular potassium must be elevated above physiological levels (to 7–9 mM) to observe regular rhythmic respiratory motor output in the hypoglossal nerve to which the preBötC projects. Reexamination of how extracellular $K^+$ affects preBötC neuronal activity has revealed that low-amplitude oscillations persist at physiological levels. These oscillatory events are subthreshold from the standpoint of transmission to motor output and are dubbed burstlets. Burstlets arise from synchronized neural activity in a rhythmogenic neuronal subpopulation within the preBötC that in some instances may fail to recruit the larger network events, or bursts, required to generate motor output. The fraction of subthreshold preBötC oscillatory events (burstlet fraction) decreases sigmoidally with increasing extracellular potassium. These observations underlie the burstlet theory of respiratory rhythm generation. Experimental and computational studies have suggested that recruitment of the non-rhythmogenic component of the preBötC population requires intracellular $Ca^{2+}$ dynamics and activation of a calcium-activated nonselective cationic current. In this computational study, we show how intracellular calcium dynamics driven by synaptically triggered $Ca^{2+}$ influx as well as $Ca^{2+}$ release/uptake by the endoplasmic reticulum in conjunction with a calcium-activated nonselective cationic current can reproduce and offer an explanation for many of the key properties associated with the burstlet theory of respiratory rhythm generation. Altogether, our modeling work provides a mechanistic basis that can unify a wide range of experimental findings on rhythm generation and motor output recruitment in the preBötC.

**\*For correspondence:**
ryan.phillips@seattlechildrens.
org (RSP);
jonrubin@pitt.edu (JER)

**Competing interest:** The authors declare that no competing interests exist.

## Editor's evaluation

This article is of significant interest to readers in the field of neural control of breathing and for researchers interested in the generation of neuronal rhythms in general. The study assembles a sophisticated computational modeling approach to test long-standing theories and emerging views in neural control of breathing and more specifically on biophysical mechanisms of burstlet generation in the respiratory network (the preBötzinger complex network). This work is an important contribution to a better understanding of the respiratory rhythm generation, will help validate (or not) running hypotheses and will guide future experiments.

## Introduction

The complex neurological rhythms produced by central pattern generators (CPGs) underlie numerous behaviors in healthy and pathological states. These activity patterns also serve as relatively experimentally accessible instances of the broader class of rhythmic processes associated with brain function. As such, CPGs have been extensively studied using a combination of experimental and computational approaches. The inspiratory CPG located in the preBötzinger complex (preBötC) in the mammalian respiratory brainstem is perhaps one of the most intensively investigated CPGs. Despite decades of research, the mechanisms of rhythm and pattern generation within this circuit remain unresolved and highly controversial; however, it appears that the pieces may now be in place to resolve this controversy.

Much of the debate in contemporary research into the mechanisms of preBötC rhythm and pattern generation revolves around the roles of specific ion currents, such as $I_{NaP}$ and $I_{CAN}$ (***Thoby-Brisson and Ramirez, 2001***; ***Del Negro et al., 2002a***; ***Koizumi and Smith, 2008***; ***Koizumi et al., 2018***; ***Picardo et al., 2019***), and whether the observed rhythm is driven by an emergent network process (***Rekling and Feldman, 1998***; ***Del Negro et al., 2005***; ***Del Negro et al., 2002b***; ***Del Negro et al., 2002b***; ***Rubin et al., 2009***; ***Sun et al., 2019***; ***Ashhad and Feldman, 2020***) and/or by intrinsically rhythmic or pacemaker neurons (***Johnson et al., 1994***; ***Koshiya and Smith, 1999***; ***Peña et al., 2004***). This debate is fueled by seemingly contradictory pharmacological blocking studies (***Del Negro et al., 2002a***; ***Peña et al., 2004***; ***Del Negro et al., 2005***; ***Pace et al., 2007b***; ***Koizumi and Smith, 2008***) and by new experimental studies (***Kam et al., 2013a***; ***Feldman and Kam, 2015***; ***Kallurkar et al., 2020***; ***Sun et al., 2019***; ***Ashhad and Feldman, 2020***) that challenge existing conceptual and computational models about the generation of activity patterns in the preBötC and underlie the so-called *burstlet theory* of respiratory rhythm generation.

A simple but reasonable hypothesis would be that a group of dedicated preBötC neurons produces a rhythmic output that induces inspiratory movement of the diaphragm, with the strength of that output tuned by some combination of the intensity of firing of these neurons and the number of neurons that become active. The conceptual framework of burstlet theory posits a more complicated two-stage view: first, inspiratory oscillations arise from an emergent, repetitive network process in a specific preBötC subpopulation dedicated to rhythm generation. These oscillations can continue independent of their downstream impact. Second, for inspiration to occur on a particular oscillation cycle, this initial activity must recruit a secondary pattern-generating subpopulation to magnify the oscillation into a full network burst capable of eliciting motor output. This hypothesis is supported by experimental preparations that compared local preBötC neuronal activity and motor output at the hypoglossal (XII) nerve in medullary slices. These studies found that in a low excitability state (controlled by the bath $K^+$ concentration, $K_{bath}$), the preBötC generates a regular rhythm featuring a mixture of large and small amplitude network oscillations, dubbed *bursts* and *burstlets*, respectively, with only the bursts eliciting XII motor output (***Kam et al., 2013a***). Moreover, the fraction of low-amplitude preBötC events (burstlet fraction) sigmoidally decreases with increasing $K_{bath}$ and only a subset of preBötC neurons are active during burstlets (***Kallurkar et al., 2020***). Importantly, preBötC bursts can be blocked by application of cadmium ($Cd^{2+}$), a calcium channel blocker, without affecting the ongoing burstlet rhythm (***Kam et al., 2013a***; ***Sun et al., 2019***), supporting the idea that rhythm generation occurs in a distinct preBötC subpopulation from pattern generation and demonstrating that conversion of a burstlet into a burst is a $Ca^{2+}$-dependent process. Finally, rhythm generation in the burstlet population is hypothesized to result from an emergent network percolation process. This last idea was developed to explain holographic photostimulation experiments, which found that optically stimulating small subsets (4–9) of preBötC inspiratory neurons were sufficient to reliably evoke endogenous-like XII inspiratory bursts with delays averaging $255 \pm 45\,\mathrm{ms}$ (***Kam et al., 2013b***). The small number of neurons required to evoke a network burst superficially seems to be at odds with reported sparse connectivities among preBötC neurons (***Rekling et al., 2000***), while models that can capture this effect via fast threshold modulation (***Rubin and Terman, 2002***) or the presentation of multiple stimulus pulses in a model of network bursting driven by synaptic dynamics (***Guerrier et al., 2015***) do not produce such extended delay durations. Additionally, these delays are on a similar timescale to the ramping pre-inspiratory neuronal activity that precedes network-wide inspiratory bursts, leading to the hypothesis that stimulation of this small set of preBötC neurons kicks off an endogenous neuronal percolation process underlying

rhythm generation, which could be initiated by the near-coincident spontaneous spiking of a small number of preBötC neurons.

The experimental underpinning of burstlet theory challenges current ideas about inspiratory rhythm and pattern generation. However, the proposed mechanisms of burst and burstlet generation remain hypothetical and, to date, there has not been a quantitative model that provides a unified, mechanistic explanation for the key experimental observations or that validates the conceptual basis for this theory. Interestingly, key components of burstlet theory, namely, that inspiratory rhythm and pattern are separable processes and that large amplitude network-wide bursts depend on calcium-dependent mechanisms, are supported by recent experimental and computational studies. Specifically, *Koizumi et al., 2018* and *Picardo et al., 2019* showed that the amplitude (i.e., pattern) of preBötC and XII bursts is controlled, independently from the ongoing rhythm, by the transient receptor potential channel (TRPM4), a calcium-activated nonselective cation current ($I_{CAN}$). These findings are consistent with burstlet theory as they demonstrate that rhythm and pattern are separable processes at the level of the preBötC. Moreover, these experimental observations are robustly reproduced by a recent computational modeling study (*Phillips et al., 2019a*), which shows that pattern generation can occur independently of rhythm generation. Consistent with burstlet theory, this model predicts that rhythm generation arises from a small subset of preBötC neurons, which in this model form a persistent sodium ($I_{NaP}$)-dependent rhythmogenic kernel, and that rhythmic synaptic drive from these neurons triggers postsynaptic calcium transients, $I_{CAN}$ activation, and amplification of the inspiratory drive potential, which drives bursting in the rest of the network.

These recent results suggest that conversion of burstlets into bursts may be $Ca^{2+}$ and $I_{CAN}$ dependent, occurring when synaptically triggered calcium transients in non-rhythmogenic preBötC neurons are intermittently large enough for $I_{CAN}$ activation to occur and to yield recruitment of these neurons into the network oscillation. The biophysical mechanism responsible for periodic amplification of $Ca^{2+}$ transients is not known, however. In this computational study, we put together and build upon these previous findings to show that periodic amplification of synaptically triggered $I_{CAN}$ transients by calcium-induced calcium release (CICR) from intracellular stores provides a plausible mechanism that can produce the observed conversion of burstlets into bursts and can explain diverse experimental findings associated with this process. Altogether, our modeling work suggests a plausible mechanistic basis for the conceptual framework of burstlet theory and the experimental observations that this theory seeks to address.

## Results

### CICR periodically amplifies intracellular calcium transients

Our first aim in this work was to test whether CICR from endoplasmic reticulum (ER) stores could repetitively amplify synaptically triggered $Ca^{2+}$ transients. To address this aim, we constructed a cellular model that includes the ER. The model features a $Ca^{2+}$ pump, which extrudes $Ca^{2+}$ from the intracellular space, a sarcoendoplasmic reticulum calcium transport ATPase (SERCA) pump, which pumps $Ca^{2+}$ from the intracellular space into the ER, and the $Ca^{2+}$-activated inositol trisphosphate (IP3) receptor (*Figure 1A*). To simulate calcium transients synaptically generated from a rhythmogenic source (i.e., burstlets), we imposed a square wave $Ca^{2+}$ current into the intracellular space with varied frequency and amplitude but fixed duration (250 ms) and monitored the resulting intracellular $Ca^{2+}$ transients. Depending on parameter values used, we observed various combinations of low- and high-amplitude $Ca^{2+}$ responses and characterized how the fraction of $Ca^{2+}$ transients that have low amplitude depends on values selected within the 2D parameter space parameterized by $Ca^{2+}$ pulse frequency and amplitude. We found that the fraction of low-amplitude $Ca^{2+}$ transients decreases as either or both of the $Ca^{2+}$ pulse frequency and amplitude are increased (*Figure 1B* and example traces C1–C4).

### Bursts and burstlets in a two-neuron preBötC network

Next, we tested whether the CICR mechanism (*Figure 1*) could drive intermittent recruitment in a reciprocally connected two-neuron network that includes one intrinsically rhythmic and one nonrhythmic neuron as a preliminary step towards considering the rhythm and pattern-generating subpopulations of the preBötC suggested by burstlet theory (*Kam et al., 2013a*; *Cui et al., 2016*; *Kallurkar et al., 2020*; *Ashhad and Feldman, 2020*) and recent computational investigation (*Phillips et al., 2019a*). In

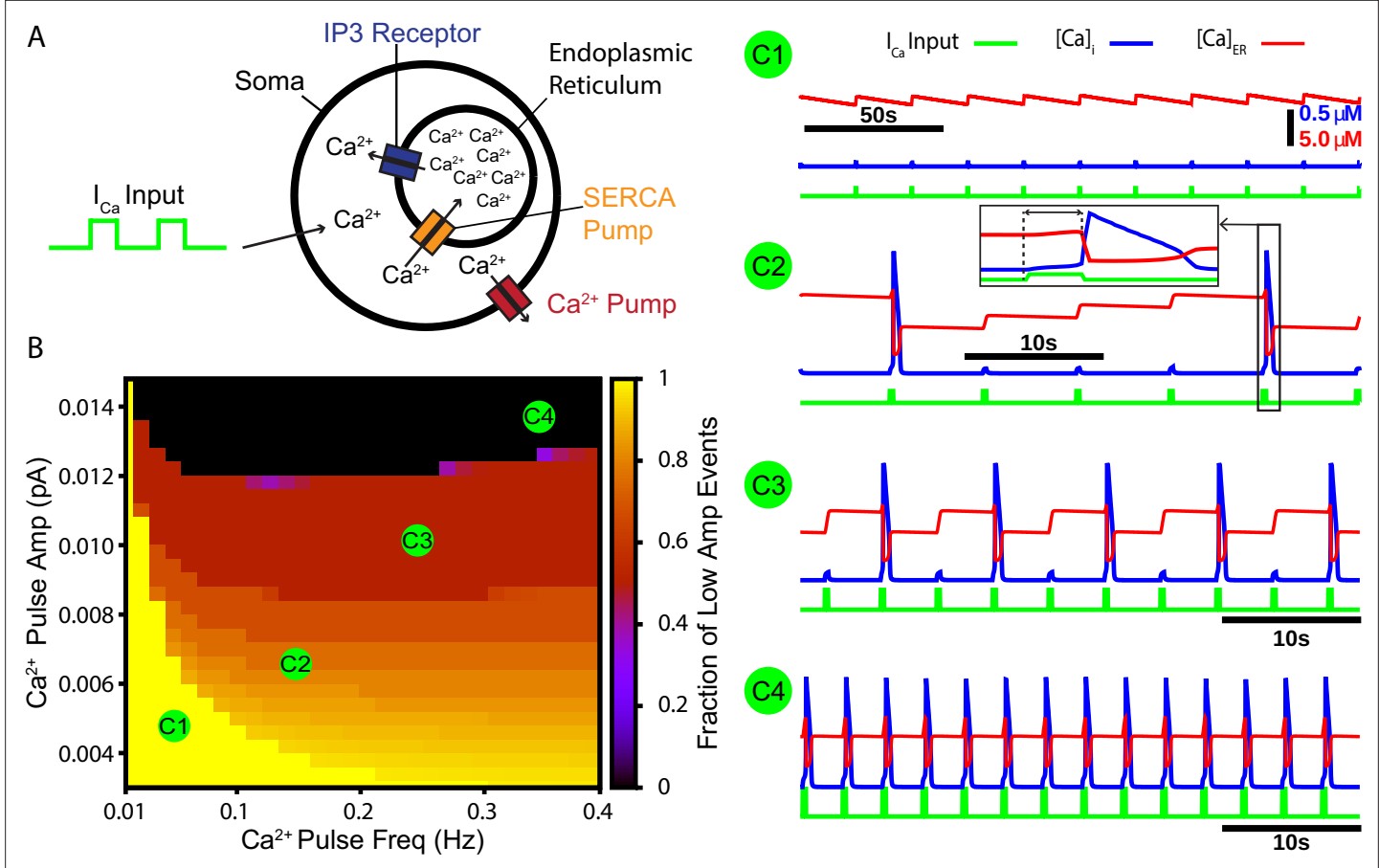

**Figure 1.** A periodic input in the form of a calcium current drives intermittent calcium-induced calcium release (CICR) from endoplasmic reticulum (ER) stores. (**A**) Schematic diagram of the model setup showing square wave profile of Ca²⁺ current input into the intracellular space, uptake of Ca²⁺ into the ER by the sarcoendoplasmic reticulum calcium transport ATPase (SERCA) pump, Ca²⁺ release through the IP3 receptor, and extrusion of Ca²⁺ through a pump in the cell membrane. (**B**) Fraction of low-amplitude intracellular Ca²⁺ transients as a function of the Ca²⁺ pulse frequency and amplitude. Pulse duration was fixed at 250 ms. (**C1–C4**) Example traces showing several ratios of low- and high-amplitude Ca²⁺ transients and the dynamics of the ER stores Ca²⁺ concentration. Inset in **C2** highlights the delay between pulse onset and CICR. The pulse amplitude and frequency for each trace are indicated in panel (**B**).

The online version of this article includes the following source data for figure 1:

**Source data 1.** Calcium-induced calcium release.

this network, neuron 1 is an $I_{NaP}$-dependent intrinsically bursting neuron, with a burst frequency that is varied by injecting an applied current, $I_{APP}$ (**Figure 2A2–A3**). The rhythmic bursting from neuron 1 generates periodic postsynaptic currents ($I_{Syn}$) in neuron 2, carried in part by Ca²⁺ ions, which are analogous to the square wave Ca²⁺ current in **Figure 1**. The amplitude of the postsynaptic Ca²⁺ transient is determined by the number of spikes per burst (**Figure 2A4**) and by the parameter $P_{SynCa}$, which determines the percentage of $I_{Syn}$ carried by Ca²⁺ ions (see 'Materials and methods' for a full description of these model components). Conversion of a burstlet (isolated neuron 1 burst) into a network burst (recruitment of neuron 2) is dependent on CICR (see **Figure 2—figure supplement 1**), which increases intracellular calcium above the threshold for $I_{CAN}$ activation.

In the reciprocally connected network, we first quantified the dependence of the burstlet fraction, which was defined as the number of burstlets (neuron 1 bursts without recruitment of neuron 2) divided by the total number of burstlets and network bursts (bursts in neuron 1 with recruitment of neuron 2), on $I_{APP}$ and $P_{SynCa}$. Increasing $I_{APP}$ increases the burst frequency in neuron 1 and decreases the number of spikes per neuron 1 burst (**Figure 2A3 and A4**), consistent with past literature (**Butera et al., 1999**; **Del Negro et al., 2001**). These changes do not strongly impact the burstlet fraction until $I_{APP}$ grows enough, at which point the shorter, more rapid bursts of neuron 1 become less effective

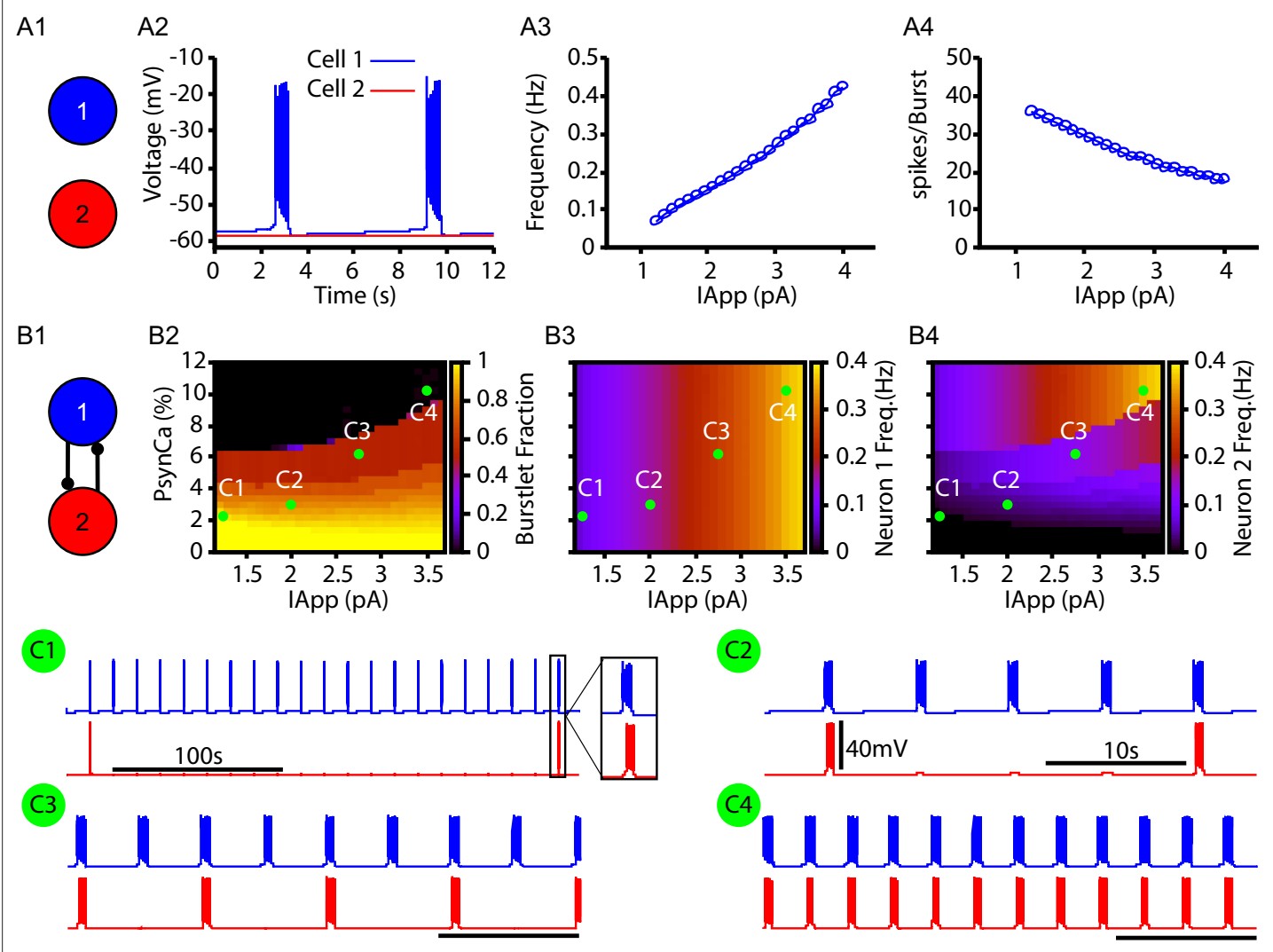

**Figure 2.** Bursts and burstlets in a two-neuron preBötzinger complex (preBötC) network. (**A1**) Schematic diagram of the synaptically uncoupled network. The rhythm- and pattern-generating components of the network are represented by neurons 1 and 2, respectively. (**A2**) Example trace showing intrinsic bursting in neuron 1 and quiescence in neuron 2. (**A3**) Burst frequency and (**A4**) the number of spikes per burst in neuron 1 as a function of an applied current ($I_{APP}$). Neuron 2 remained quiescent within this range of $I_{APP}$. (**B1**) Schematic diagram of the synaptically coupled network. (**B2–B4**) 2D plots characterizing the (**B2**) burstlet fraction, (**B3**) neuron 2 (burst) frequency, and (**B4**) neuron 2 spikes per burst (burst amplitude) as a function of $I_{APP}$ and $P_{SynCa}$. (**C1–C4**) Example traces for neurons 1 and 2 for various $I_{APP}$ and $P_{SynCa}$ values indicated in (**B2–B4**). Notice the scale bar is 100s in **C1** and 10s in (**C2–C4**). Inset in (**C1**) shows the burst shape not visible on the 100 s timescale. The model parameters used in these simulations are: (neurons 1 and 2) $K_{Bath} = 8\,\text{mM}$, $g_{Leak} = 3.35\,\text{nS}$, $W_{12} = W_{21} = 0.006\,\text{nS}$; (neuron 1) $g_{NaP} = 3.33\,\text{nS}$, $g_{CAN} = 0.0\,\text{nS}$, (neuron 2) $g_{NaP} = 1.5\,\text{nS}$, $g_{CAN} = 1.5\,\text{nS}$.

The online version of this article includes the following source data and figure supplement(s) for figure 2:

**Source data 1.** Burstlets and bursts in a two-neuron network.

**Figure supplement 1.** Without calcium-induced calcium release (CICR), the two-neuron network fails to generate bursts (recruitment of neuron 2).

at recruiting neuron 2 and thus the burstlet fraction increases (***Figure 2B2***; note that increasing $I_{APP}$ corresponds to a horizontal cut through the panel). In general, increasing $P_{SynCa}$ decreased the burstlet fraction (i.e., increased the frequency of neuron 2 recruitment) by causing a larger calcium influx with each neuron 1 burst (see ***Figure 2B2 and C1–C4***).

The burst frequency in neuron 2 is determined by the burst frequency of neuron 1 and the burstlet fraction. These effects determine the impact of changes in $P_{SynCa}$ and $I_{APP}$ on neuron 2 burst frequency (***Figure 2B3***). As $I_{APP}$ increases, the rise in burstlet frequency implies that neuron 2 bursts in response to a smaller fraction of neuron 1 bursts, yet the rise in neuron 1 burst frequency means that these

bursts occur faster. These two effects can balance to yield a relatively constant neuron 2 frequency, although the balance is imperfect and frequency does eventually increase. Increases in $P_{SynCa}$ more straightforwardly lead to increases in neuron 2 burst frequency as the burstlet fraction drops.

Finally, the number of spikes per burst in neuron 2 is not strongly affected by changes in $I_{APP}$ and $P_{SynCa}$ (*Figure 2B4*), suggesting an all-or-none nature of recruitment of bursting in neuron 2. Interestingly, the period between network bursts (i.e., time between neuron 2 recruitment events) can be on the order of hundreds of seconds (e.g., *Figure 2C1*). This delay is consistent with some of the longer timescales shown in experiments characterizing bursts and burstlets (*Kallurkar et al., 2020*).

## CICR supports bustlets and bursts in a data-constrained preBötC network model

Next, we tested whether the CICR mechanism presented in *Figures 1 and 2* could underlie the conversion of burstlets into bursts in a larger preBötC model network including rhythm- and pattern-generating subpopulations (see 'Data analysis and definitions' section for details on how these are distinguished in the network setting) and whether this network could capture the $K_{bath}$-dependent changes in the burstlet fraction characterized in *Kallurkar et al., 2020*. $K_{bath}$ sets the extracellular K$^+$

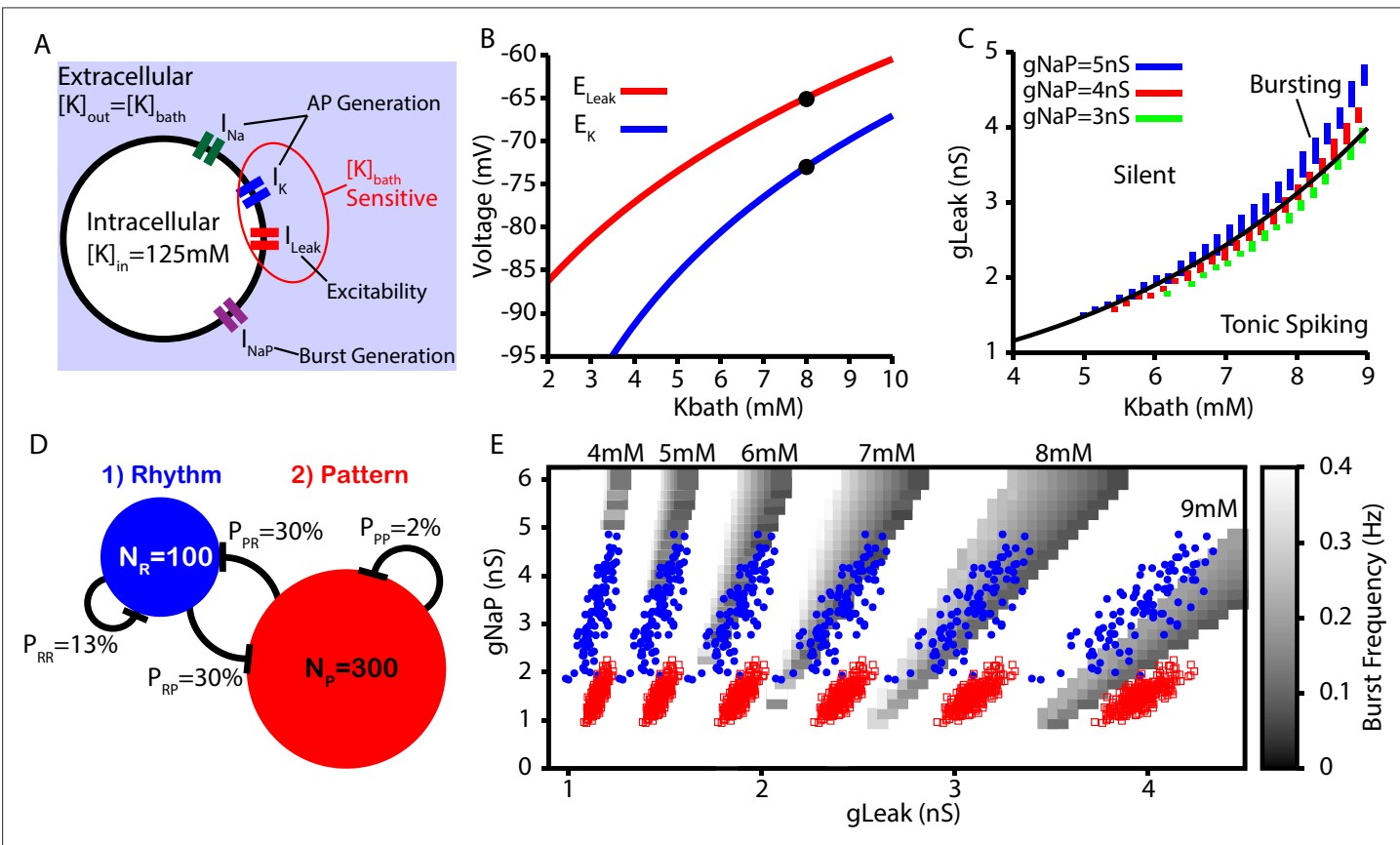

**Figure 3.** Intrinsic cellular and network dynamics depend on the bath potassium concentration. (**A**) Schematic diagram of an isolated model preBötzinger complex (preBötC) neuron showing the simulated ion channels involved in AP generation, excitability, and burst generation, as well as indication of currents directly affected by changing the bath potassium concentration ($K_{bath}$). (**B**) Dependence of potassium ($E_K$) and leak ($E_{Leak}$) reversal potentials on $K_{bath}$. Black dots indicate experimentally measured values for $E_K$ and $E_{Leak}$ from *Koizumi and Smith, 2008*. (**C**) Dependence of intrinsic cellular dynamics on $K_{bath}$, $g_{Leak}$, and $g_{NaP}$. Black curve represents the relationship between $K_{Bath}$ and $g_{Leak}$ used in the full preBötC network. (**D**) Schematic diagram of size and connectivity probabilities of the rhythm- and pattern-generating populations within the preBötC model. (**E**) 2D plot between $g_{NaP}$ and $g_{Leak}$ showing the location of the intrinsic bursting regime for varied concentrations of $K_{Bath}$. The distributions of neuronal conductances in the rhythm- and the pattern-generating populations are indicated by the blue dots and red squares, respectively.

The online version of this article includes the following source data and figure supplement(s) for figure 3:

**Source data 1.** Bath potassium concentration dependence of cellular and network dynamics.

**Figure supplement 1.** Dependence of intrinsic cellular dynamics and the number of spikes per burst on $K_{bath}$ and $g_{Leak}$.

concentration, which in turn determines the driving force for any ionic currents that flux $K^+$. In preBötC neurons, these currents include the fast $K^+$ current, which is involved in action potential generation, and the $K^+$-dominated leak conductance, which primarily affects excitability (**Figure 3A**). In our simulations, we modeled the potassium ($E_K$) and leak ($E_{Leak}$) reversal potentials as functions of $K_{bath}$ using the Nernst and Goldman–Hodgkin–Katz equations. The resulting curves were tuned to match existing data from **Koizumi and Smith, 2008**, as shown in **Figure 3B**. In our simulations, we found that intrinsic bursting is extremely sensitive to changes in $K_{bath}$. However, with increasing $K_{bath}$, intrinsic bursting could be maintained over a wide range of $K^+$ concentrations when accompanied by increases in $g_{Leak}$ (**Figure 3C**). Additionally, the number of spikes per burst in the bursting regime increases with $K_{bath}$ (**Figure 3—figure supplement 1**). This $K_{bath}$-dependence of $g_{Leak}$ is consistent with experimental data showing that neuronal input resistance decreases with increasing $K_{bath}$ (**Okada et al., 2005**).

To construct a model preBötC network, we linked rhythm- and pattern-generating subpopulations via excitatory synaptic connections within and between the two populations (**Figure 3D**). We distinguished the two populations by endowing them with distinct distributions of persistent sodium current conductance ($g_{NaP}$), as documented experimentally (**Del Negro et al., 2002a**; **Koizumi and Smith, 2008**). In both populations, we maintained the dependence of $g_{Leak}$ on $K_{bath}$ (see **Figure 3C and E**).

For the full preBötC network model, we first characterized the impact of changes in $K_{bath}$ on network behavior without calcium dynamics by setting $P_{SynCa} = 0$. This network condition is analogous to in vitro preparations where all $Ca^{2+}$ currents are blocked by $Cd^{2+}$ and the preBötC can only generate burstlets (**Kam et al., 2013a**; **Sun et al., 2019**). Not surprisingly, with calcium dynamics blocked, we found that the network can only generate small amplitude network oscillations (burstlets) that first emerge at approximately $K_{bath} = 5\,\text{mM}$ (**Figure 4A**). Moreover, under these conditions, increasing $K_{bath}$ results in an increase in the burstlet frequency and amplitude (**Figure 4B and C**), which is consistent with experimental observations (**Kallurkar et al., 2020**).

In the full network with calcium dynamics ($P_{SynCa} > 0$), burstlets generated by the rhythmogenic subpopulation will trigger postsynaptic calcium transients in the pattern-generating subpopulation. Therefore, in this set of simulations the burstlet activity of the rhythm generating population plays an analogous role to the square wave $Ca^{2+}$ current in **Figure 1** and to bursts of the intrinsically rhythmic neuron in **Figure 2**. Hence, we characterized the burstlet fraction, burst frequency, and burst amplitude – with a burst defined as an event in which a burstlet from the rhythm generating population recruits a burst in the pattern-generating population – in the full preBötC model network as a function of $K_{bath}$ and $P_{SynCa}$ (**Figure 4D–F**). In this case, the frequency of the postsynaptic $Ca^{2+}$ oscillation is controlled by $K_{bath}$ (**Figure 4B**). However, because $K_{bath}$ also affects burstlet amplitude (**Figure 4C**), the postsynaptic $Ca^{2+}$ amplitude is determined by both $K_{bath}$ and $P_{SynCa}$. If $K_{bath}$ is held fixed, then modulating $P_{SynCa}$ will only affect the amplitude of the postsynaptic $Ca^{2+}$ transient since burstlet amplitude will not be impacted. The effects of selectively changing the postsynaptic $Ca^{2+}$ amplitude on the full network can thus be extracted by considering a vertical slice through **Figure 4E–F**. Note that in the simulations that we show here burstlet generation arises from a mechanism based on $I_{NaP}$; however, we obtain similar network results if we impose burstlet activity on the burstlet subpopulation and maintain the coupling between populations and $Ca^{2+}$ dynamics for burst recruitment (**Figure 4—figure supplement 1**).

We found that increasing $P_{SynCa}$ or $K_{bath}$ generally decreases the burstlet fraction, increases burst frequency, and slightly increases the burst amplitude (**Figure 4D–F and G1–G**). The decrease in the burstlet fraction with increasing $K_{bath}$ or $P_{SynCa}$ is caused by the increase in the burstlet amplitude (**Figure 4C**) or in $Ca^{2+}$ influx with each burstlet, respectively, both of which increase the $Ca^{2+}$ transient in the pattern-generating subpopulation. The increase in burst frequency with increases in $K_{bath}$ or $P_{SynCa}$ is due to the decreased burstlet fraction (i.e., the burstlet to burst transitions occur on a greater proportion of cycles) and, in the case of $K_{bath}$, by an increase in the burstlet frequency (**Figure 4B**). The slight increase in burst amplitude with increasing $K_{bath}$ is largely due to the increase in the burstlet amplitude (**Figure 4C**). **Figure 4H** highlights the relative shape of burstlets and bursts as well as the delay between burstlet generation and recruitment of the pattern-generating population and simulated hypoglossal output, which agrees qualitatively with experimental observations (**Kallurkar et al., 2020**). Experimentally, it is likely that postsynaptic $Ca^{2+}$ transients will increase with increasing $K_{bath}$ due to the change in the resting $V_m$ in nonrhythmic preBötC neurons (**Tryba et al., 2003**) relative to

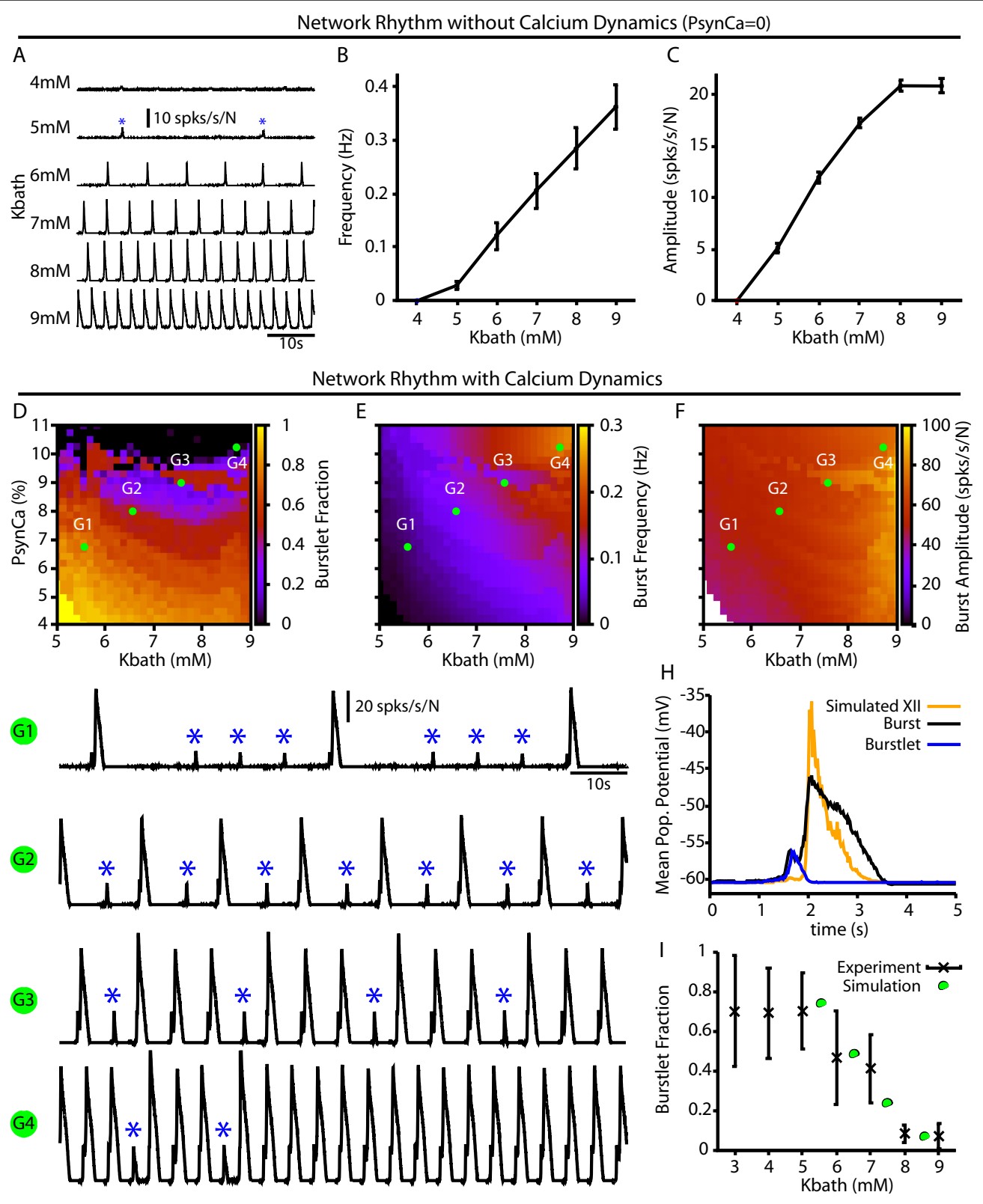

**Figure 4.** Burstlets and bursts in a 400-neuron preBötzinger complex (preBötC) network model with and without calcium dynamics. (**A**) Rhythmogenic output of the simulated network without calcium dynamics ($P_{SynCa} = 0$) as a function of $K_{Bath}$. These oscillations are considered burstlets as they are incapable of recruiting the pattern-generating population without calcium dynamics. (**B**) Frequency and (**C**) amplitude of the burstlet oscillations as a function of $K_{bath}$. (**D–F**) 2D plots characterizing the (**D**) burstlet fraction, (**E**) the burst frequency, and (**F**) the burst amplitude as a function of $K_{bath}$ and

*Figure 4 continued on next page*

*Figure 4 continued*

$P_{SynCa}$ (note that the $P_{SynCa}$ range shown does not start at 0). (**G1–G4**) Example traces illustrating a range of possible burstlet fractions generated by the network. Burstlets are indicated by asterisks. (**H**) Overlay of the average population voltage during bursts and burstlets. The hypoglossal output is calculated by passing the mean population amplitude through a sigmoid function $f = -60.5 + 60/[1 + e^{-(x+45)/2.5}]$. (**I**) Burstlet fraction as a function of $K_{bath}$ for the four example traces indicated in panels (**G1–G4**). *Figure 4I* has been adapted from Figure 1B from *Kallurkar et al., 2020*.

The online version of this article includes the following source data and figure supplement(s) for figure 4:

**Source data 1.** Burstlets and Bursts in a larger network.

**Figure supplement 1.** Burstlets and bursts in a 400-neuron preBötzinger complex (preBötC) network model with an imposed burstlet rhythm.

the voltage-gated activation dynamics of postsynaptic calcium channels (*Elsen and Ramirez, 1998*); see 'Discussion' for a full analysis of this point. Interestingly, in our simulations, increasing $P_{SynCa}$ (i.e., the amplitude of the postsynaptic calcium transients) with $K_{bath}$ (*Figure 4* traces G1–G4) generated $K_{bath}$-dependent changes in the burstlet fraction that are consistent with experimental observations (*Kallurkar et al., 2020*; see *Figure 4I*).

Note that our model includes synaptic connections from pattern-generating neurons back to rhythm-generating neurons. These connections prolong activity of rhythmic neurons in bursts, relative to burstlets, which in turn yields a longer pause before the next event (e.g., *Figure 4G1*). This effect can constrain event frequencies somewhat in the fully coupled network relative to the feedforward case (e.g., frequencies in *Figure 4B* exceed those in *Figure 4E* for comparable $K_{bath}$ levels).

## Calcium and $I_{CAN}$ block have distinct effects on the burstlet fraction

Next, we further characterized the calcium dependence of the burstlet to burst transition in our model by simulating calcium blockade or $I_{CAN}$ blockade by a progressive reduction of $P_{SynCa}$ or $g_{CAN}$, respectively. We found that complete block of synaptically triggered $Ca^{2+}$ transients or $I_{CAN}$ block eliminates bursting without affecting the underlying burstlet rhythm (*Figure 5A and B*). Interestingly, progressive blockades of each of these two mechanisms have distinct effects on the burstlet fraction: blocking postsynaptic $Ca^{2+}$ transients increases the burstlet fraction by increasing the number of burstlets required to trigger a network burst, whereas $I_{CAN}$ block only slightly increases the burstlet fraction (*Figure 5C*). In both cases, however, progressive blockade smoothly decreases the amplitude of network bursts (*Figure 5D*). The decrease in amplitude in the case of $I_{CAN}$ block is due to derecruitment of neurons from the pattern-forming subpopulation and a decrease in the firing rate of the neurons that remain active, whereas in the case of $Ca^{2+}$ block the decrease in amplitude results primarily from derecruitment (*Figure 5E and F*). These simulations provide mechanism-specific predictions that can be experimentally tested.

## Dose-dependent effects of opioids on the burstlet fraction

Recent experimental results by *Baertsch et al., 2021* showed that opioid application locally within the preBötC decreases burst frequency but also increases the burstlet fraction. In the preBötC, opioids affect neuronal dynamics by binding to the μ-opioid receptor (μOR). The exact number of preBötC neurons expressing μOR is unclear; however, the number appears to be small, with estimates ranging from 8% to 50% (*Bachmutsky et al., 2020*; *Baertsch et al., 2021*; *Kallurkar et al., 2021*). Additionally, μOR is likely to be selectively expressed on neurons involved in rhythm generation, given that opioid application in the preBötC primarily impacts burst frequency rather than amplitude (*Sun et al., 2019*; *Baertsch et al., 2021*). Importantly, within the preBötC, opioids ultimately impact network dynamics through two distinct mechanisms: (1) hyperpolarization, presumably via activation of a G protein-gated inwardly rectifying potassium leak (GIRK) current (*Kubo et al., 1993*; *Gray et al., 1999*; *Montandon et al., 2016*), and (2) decreased excitatory synaptic transmission, presumably via decreased presynaptic release (*Ballanyi et al., 2009*; *Wei and Ramirez, 2019*; *Baertsch et al., 2021*).

Taking these considerations into account, we tested if our model could explain the experimental observations. Specifically, we simulated opioids as having a direct impact only on the neurons within the rhythmogenic population and their synaptic outputs (*Figure 6A*). To understand how preBötC network dynamics are impacted by the two mechanisms through which opioids have been shown to act, we ran separate simulations featuring either activation of GIRK channels or block of the synaptic output from the rhythmogenic subpopulation (*Figure 6B–F*). We found that both GIRK activation

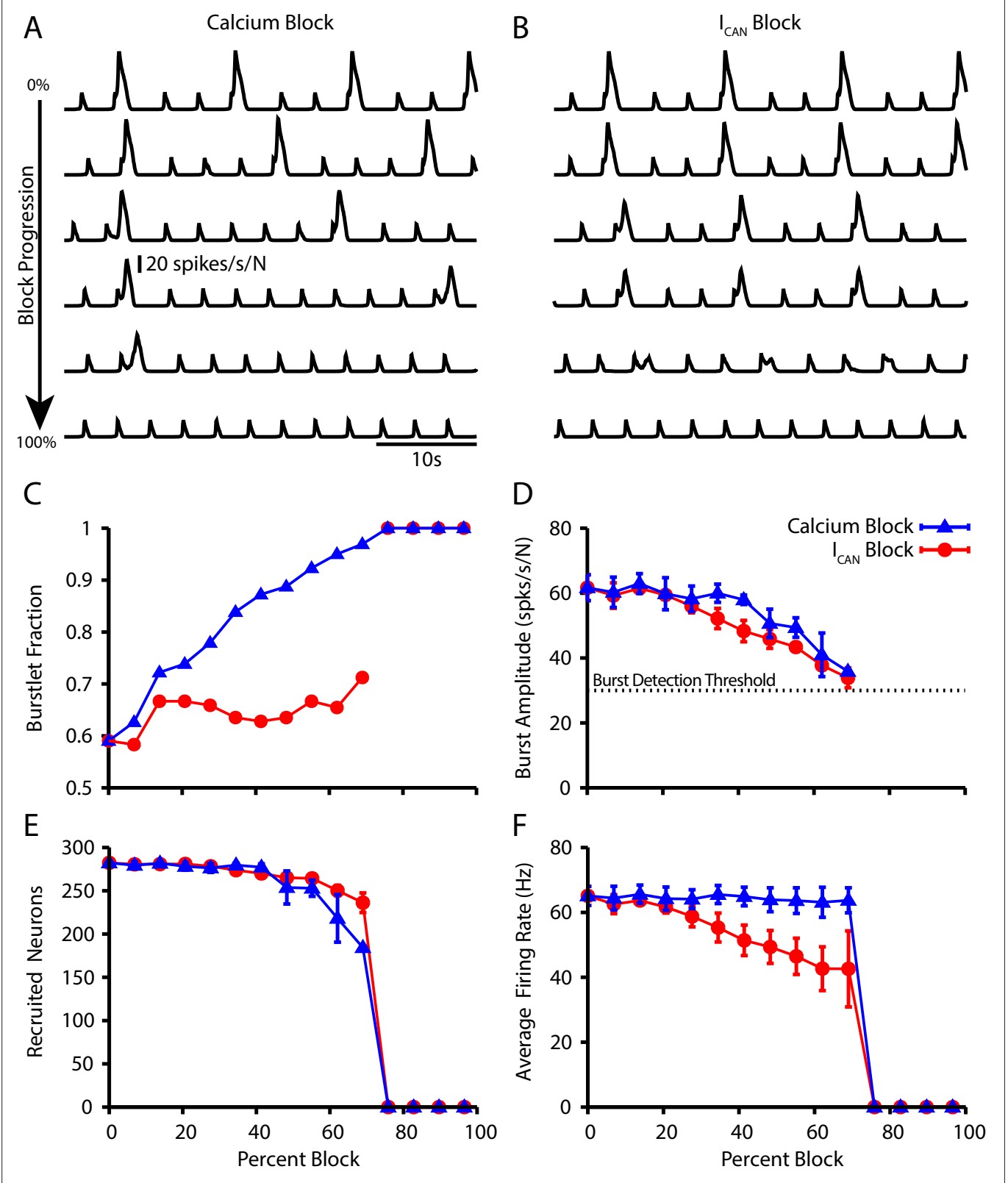

**Figure 5.** Effect of Ca²⁺ and CAN current blockade on burstlets and bursts. Network traces showing the effect of (**A**) calcium current blockade ($P_{SynCa}$ reduction) and (**B**) CAN current blockade ($g_{CAN}$ reduction) on the period and amplitude of bursts. Effects of calcium or $I_{CAN}$ blockade on (**C**) the burstlet fraction, (**D**) the amplitude of bursts and (**E**) the number of recruited and (**F**) peak firing rate of recruited neurons in pattern-generating subpopulation during network bursts as a function of the blockade percentage.

*Figure 5 continued on next page*

*Figure 5 continued*

The online version of this article includes the following source data for figure 5:

**Source data 1.** Simulated calcium or CAN current blockade.

and synaptic block reduced the burst frequency (*Figure 6D*) and slightly increased burst amplitude (*Figure 6E*). The decreased frequency with synaptic block comes from an increase in the burstlet fraction, whereas GIRK activation kept the burstlet fraction constant while reducing the burstlet frequency (*Figure 6F*). Finally, combining these effects, we observed that simultaneously increasing the GIRK channel conductance and blocking the synaptic output of μOR-expressing neurons in our simulations generates slowing of the burst frequency and an increase in the burstlet fraction consistent with in vitro experimental data (*Figure 6D–G*).

## Simultaneous stimulation of subsets of preBötC neurons elicits network bursts with long delays

Simultaneous stimulation of 4–9 preBötC neurons in in vitro slice preparations has been shown to be sufficient to elicit network bursts with similar patterns to those generated endogenously (*Kam et al., 2013b*). These elicited bursts occur with delays of several hundred milliseconds relative to the stimulation time, which is longer than would be expected from existing models. Interestingly, in the current model, due to the dynamics of CICR, there is a natural delay between the onset of burstlets and the recruitment of the follower population that underlies the transition to a burst. Therefore, we investigated whether our model could match and explain the observations seen in *Kam et al., 2013b*.

In our model, we first calibrated our stimulation to induce a pattern of spiking that is comparable to the patterns generated in *Kam et al., 2013b* (10–15 spikes with decrementing frequency, *Figure 7A*). We found that stimulation of 3–9 randomly selected neurons could elicit network bursts with delays on the order of hundreds of milliseconds (*Figure 7B and C*). Next, we characterized (1) the probability of eliciting a burst, (2) the delay in the onset of elicited bursts, and (3) the variability in delay, each as a function of the time of stimulation relative to the end of an endogenous burst (i.e., a burst that occurs without stimulation) and of the number of neurons stimulated (*Figure 7D–F*). In general, we found that increasing the number of stimulated neurons increases the probability of eliciting a burst and decreases the delay between stimulation and burst onset. Moreover, the probability of eliciting a burst increases and the delay decreases as the time after an endogenous burst increases (*Figure 7G and H*). Additionally, with its baseline parameter tuning, our model had a refractory period of approximately 1 s following an endogenous burst during which stimulation could not evoke a burst (*Figure 7*). The refractory period in our model is longer than measured experimentally (500 ms) (*Kam et al., 2013b*).

To determine the mechanisms involved in the refractoriness, we plotted the time courses of key slow variables in the model, namely, persistent sodium inactivation $h_{NaP}$, ER calcium ($[Ca]_{ER}$), and synaptic depression $D$, over one burst cycle in the absence of stimulation (see *Figure 7—figure supplement 1*). We found that the recovery from synaptic depression and the deinactivation of $h_{NaP}$ were the two slow processes with time courses that aligned with the loss of refractoriness. Thus, in our model, it appears that these two factors are crucial to the probability that a stimulus will elicit a sustained response, while calcium-related effects predominantly relate to the recruitment process by which such a response develops into a burst.

To conclude our investigation, we examined how changes in the connection probability within the pattern-forming population ($P_{PP}$) affect the refractory period, probability, and delay of evoked bursts following simultaneous stimulation of 3–9 randomly selected neurons in the preBötC population. We focused on the pattern-forming population because it comprises 75% of the preBötC population, and, therefore, neurons from this population are most likely to be stimulated and the synaptic projections from these neurons are most likely to impact the properties of evoked bursts. To avoid a confound that would arise if increased connection probability led to overall stronger synaptic input, we adjusted $W_{PP}$ to compensate for changes in $P_{PP}$ and keep the network synaptic strength, defined as $S = N_P \cdot P_{PP} \cdot W_{PP}$, at a constant value.

With this scaling, we found that decreasing/increasing $P_{PP}$ decreased/increased the refractory period (*Figure 8A–C*) by impacting the probability of eliciting a burst in the period immediately after an endogenous burst (*Figure 8D and E*). More specifically, the change in the probability of evoking

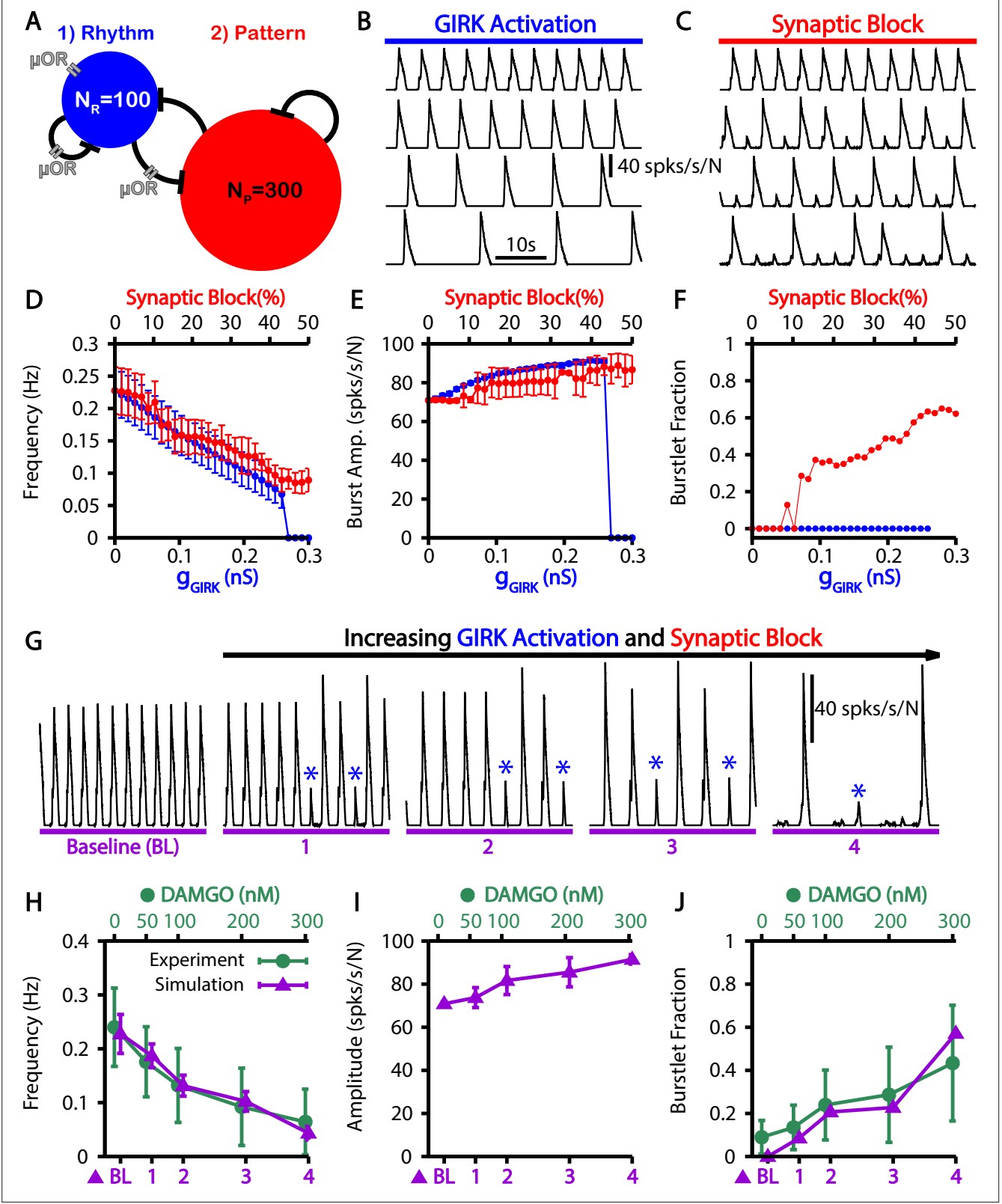

**Figure 6.** Simulated μ-opioid receptor (μOR) activation by local DAMGO application in the preBötzinger complex (preBötC) and comparison with experimental data. (**A**) Schematic preBötC network diagram showing the location of μOR. Example traces showing the effect of progressive (from top to bottom) (**B**) $g_{GIRK}$ channel activation and (**C**) synaptic block on the network output. Quantification of $g_{GIRK}$ activation or synaptic block by μOR on the (**D**) burst frequency, (**E**) burst amplitude, and (**F**) burstlet fraction. Error bars indicate SD. (**G**) Example traces showing the effects of progressive increases

*Figure 6 continued on next page*

*Figure 6 continued*

in $g_{GIRK}$ and synaptic block on network output. Burstlets are indicated by blue asterisks. The parameters for each case are as follows: (BL) $g_{GIRK} = 0.0\,\text{nS}$, $\gamma_{\mu OR} = 0.0$; (1) $g_{GIRK} = 0.031034\,\text{nS}$, $\gamma_{\mu OR} = 0.81034$; (2) $g_{GIRK} = 0.093103\,\text{nS}$, $\gamma_{\mu OR} = 0.7069$; (3) $g_{GIRK} = 0.14483\,\text{nS}$, $\gamma_{\mu OR} = 0.68966$; (4) $g_{GIRK} = 0.19655\,\text{nS}$, $\gamma_{\mu OR} = 0.58621$. Comparison of experimental data and the effects of progressive increases in $g_{GIRK}$ and synaptic block on the (H) frequency and (I) amplitude of bursts as well as (J) the burstlet fractions for the traces shown in (G). *Figure 6H and J* have been adapted from Figure 3C and E from *Baertsch et al., 2021*. The effects of DAMGO on burst amplitude were not quantified in *Baertsch et al., 2021*.

The online version of this article includes the following source data for figure 6:

**Source data 1.** Effects of simulated opioids on burstlets and bursts.

a burst, with decreased/increased $P_{PP}$, is indicated by a leftward/rightward shift in the probability vs. stimulation time curves relative to a control level of $P_{PP}$ ($P_{PP} = 2\%$) (see *Figure 8D and E*). That is, relatively small connection probabilities with large connection strengths lead to network dynamics with a shorter refractory period when stimulation cannot elicit a burst and a higher probability that a stimulation at a fixed time since the last burst will evoke a new burst.

It may seem surprising that networks with smaller connection probabilities exhibit a faster emergence of bursting despite their larger connection weights since intuitively, with lower connection probabilities, fewer neurons could be recruited by each action potential, resulting in longer, more time-consuming activation pathways. A key point, however, is that when connection weights are larger, fewer temporally overlapping inputs are needed to recruit each inactive neuron. For example, suppose that we fix $N_P$ and $W_{PP}$, and we take $P_{PP}$ to scale as $1/N_P$. The minimal number of inputs from active neurons needed to activate an inactive neuron depends on the synaptic weight, $W_{PP}$. Let $r$ denote this number for the specific value of $W_{PP}$ that we have selected. We can approximate the expected number of neurons receiving $r$ or more inputs from $A$ active neurons by computing the expected number receiving exactly $r$ inputs, which we denote as $[I_r]$, where the brackets indicate an expectation or average. For a network with a random connectivity profile, this expected value is computed from the binomial formula as

$$[\mathrm{I}_r] = \binom{A}{r} \left(\frac{1}{N_P}\right)^r \left(1 - \frac{1}{N_P}\right)^{A-r}.$$

Suppose that next we consider another network in which we double $P_{PP}$ and halve $W_{PP}$, thus keeping their product constant. For this smaller $W_{PP}$, more inputs will be needed to activate an inactive neuron. Specifically, assume that now at least $2r$ inputs are needed for activation. The expected number of neurons receiving $2r$ inputs, $[I_{2r}]$, is given by

$$[\mathrm{I}_{2r}] = \binom{A}{2r} \left(\frac{2}{N_P}\right)^{2r} \left(1 - \frac{2}{N_P}\right)^{A-2r}.$$

An elementary calculation shows that $[I_{2r}] < [I_r]$ for relevant parameter values (such as $N_P = 300$ and small $r$ as indicated by the stimulation experiments). Thus, increasing $P_{PP}$ and proportionally scaling down $W_{PP}$ reduces the chance of successful recruitment of inactive neurons by active neurons.

Interestingly, our simulations suggest that the connection probability in the pattern-generating population must be between 1% and 2% to match the approximately 500 ms refractory period measured experimentally (*Kam et al., 2013b*; *Figure 8F*). Surprisingly, the mean and distribution of delays from stimulation to burst for all successfully elicited bursts are not strongly affected by changes in $P_{PP}$ (*Figure 8F*). For a given stimulation time and number of neurons stimulated, however, decreasing $P_{PP}$ decreases the delay of elicited bursts (*Figure 8G*). Finally, because the neurons in the pattern-generating population appear to play a dominant role in determining if stimulation will elicit a network burst, we characterized how the number of pattern-generating neurons stimulated, out of a total set of nine stimulated neurons, affects the probability of eliciting a network burst as a function of stimulation time (*Figure 8H*). These simulations were carried out under a baseline condition of $P_{PP} = 2\%$. In general, we found that stimulating a relatively larger proportion of pattern-generating neurons increased the probability of eliciting a network burst for all times after the approximately 1

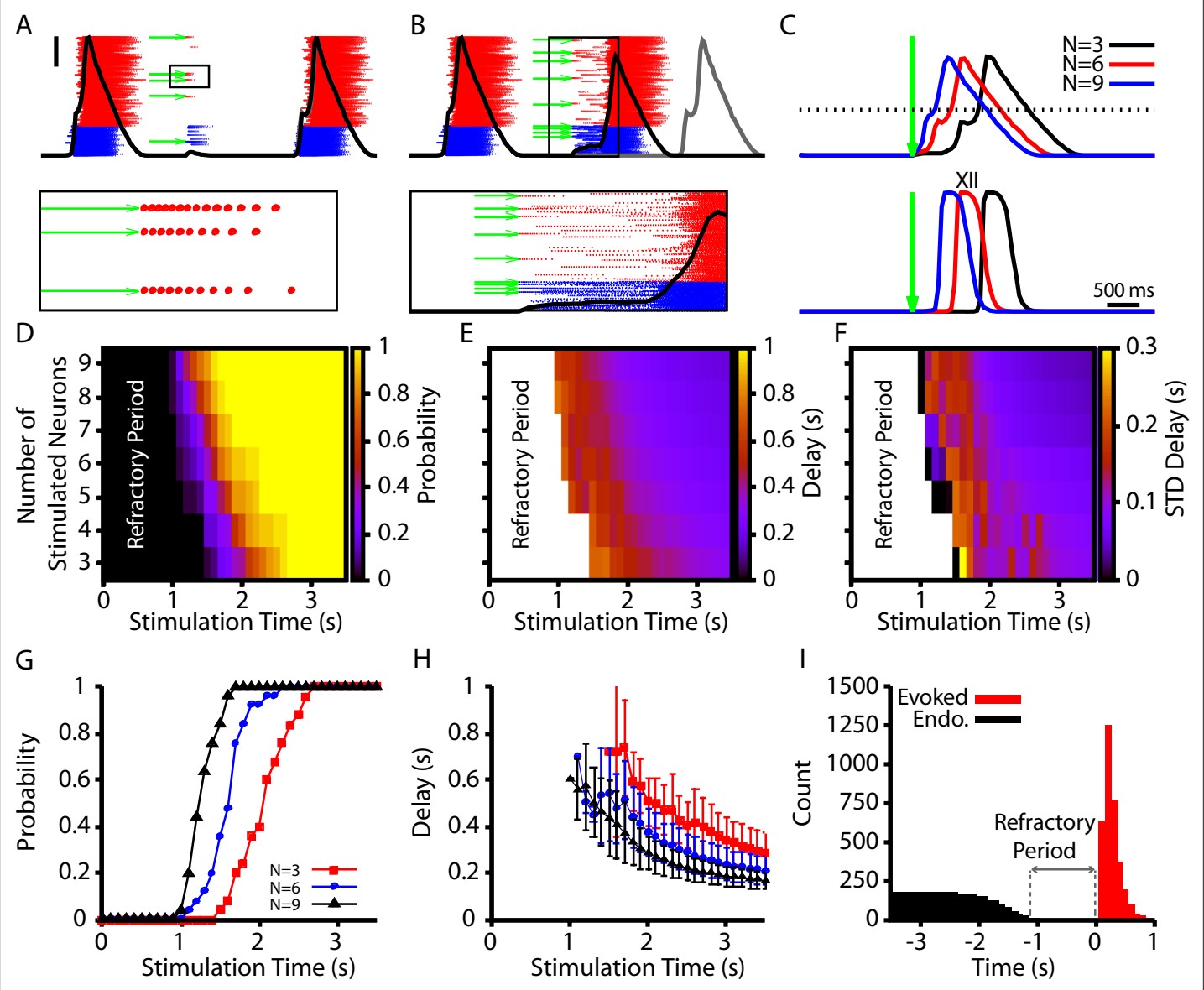

**Figure 7.** Evoked population bursts by simulated holographic stimulation of 3–9 preBötzinger complex (preBötC) neurons. (**A**) Raster plot of neuronal spiking triggered by simulated holographic stimulation of six preBötC neurons shortly after an endogenous burst and resulting failure to evoke a network burst. Black line represents the integrated population activity. Scale bar indicates 20 spikes/s/N. Bottom panel shows the spiking activity triggered in individual neurons by the simulated holographic stimulation. Panel duration is 1 s. (**B**) Example simulation where stimulation of nine preBötC neurons evokes a network burst. Gray curve indicates timing of the next network burst if the network was not stimulated. (Bottom panel) Expanded view of the percolation process that is triggered by holographic stimulation on a successful trial. Panel duration is 1.75 s. (**C**) Example traces showing the delay between the stimulation time and the evoked bursts as a function of the number of neurons stimulated for the (top) integrated preBötC spiking and (bottom) simulated hypoglossal activity. (**D–F**) Characterization of (**D**) the probability of evoking a burst, (**E**) the mean delay of evoked bursts, and (**F**) the standard deviation of the delay as a function of the time after an endogenous burst and the number of neurons stimulated. (**G**) Probability and (**H**) delay as a function of the stimulation time for stimulation of three, six, or nine neurons. Error bars in (**H**) indicate SD. (**I**) Histogram of evoked and endogenous bursts relative to the time of stimulation ($t = 0\,s$) for all successful trials in all simulations; notice a 1 s refractory period.

The online version of this article includes the following source data and figure supplement(s) for figure 7:

**Source data 1.** Simulated holographic stimulation.

**Figure supplement 1.** Dynamics of $I_{NaP}$ inactivation ($h_{NaP}$), endoplasmic reticulum (ER) calcium concentration ($[Ca]_{ER}$), and synaptic depression and recovery ($D$) as a function of time relative to the inspiratory cycle.

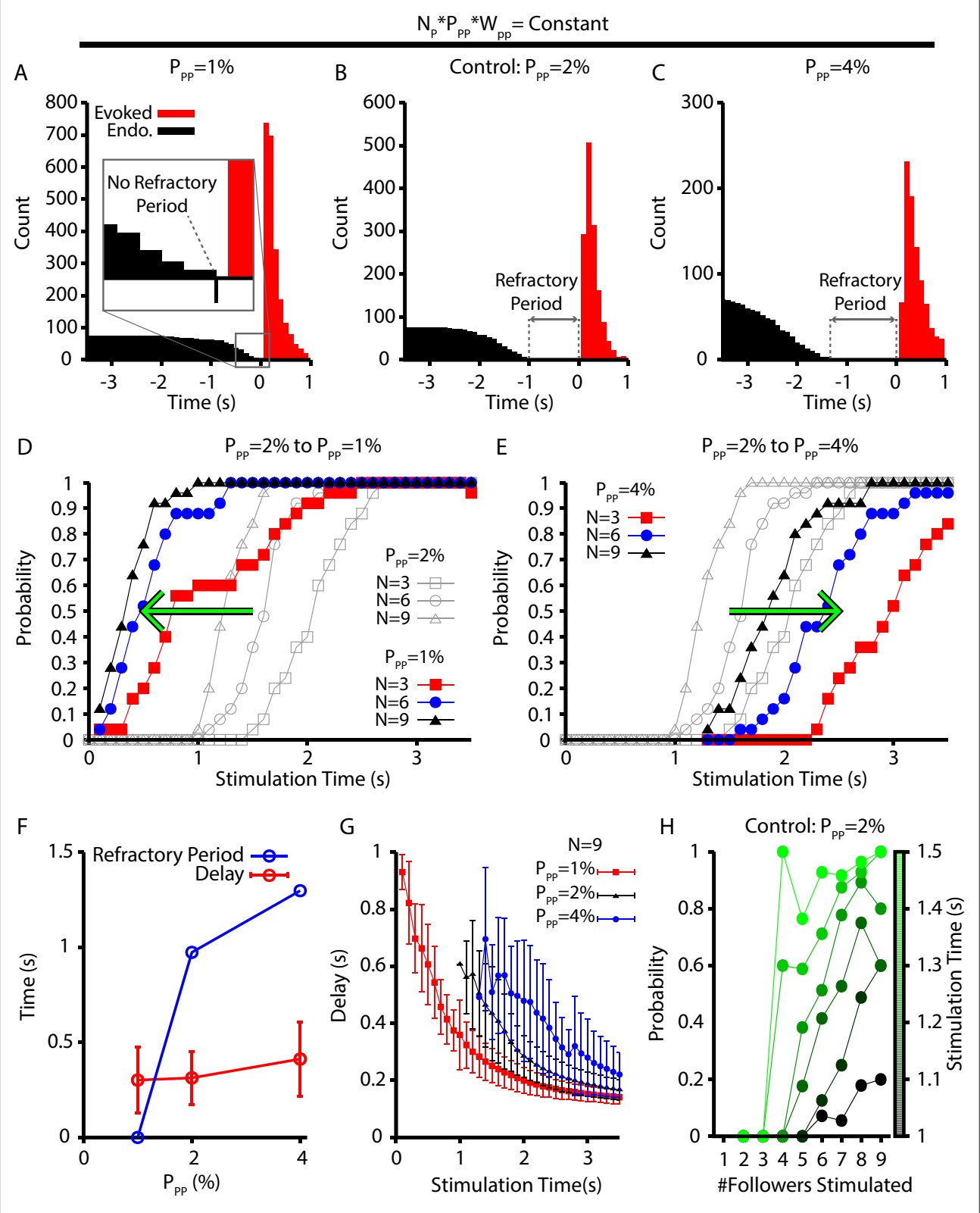

**Figure 8.** Refractory period and delay of evoked bursts following simulated holographic stimulation depend on the follower network connectivity. (**A–C**) Histogram of evoked and endogenous bursts relative to the time of stimulation ($t = 0\,\mathrm{s}$) for all successful trials where three, six, and nine neurons were stimulated and for different connection probabilities (but fixed total network synaptic strength; i.e., $N_P \cdot P_{PP} \cdot W_{PP} = constant$) in the follower population: (**A**) $P_{PP} = 1\%$; (**B**) $P_{PP} = 2\%$; and (**C**) $P_{PP} = 4\%$. (**D, E**) Effect of (**D**) decreasing ($2\% \rightarrow 1\%$) and (**E**) increasing ($2\% \rightarrow 4\%$) the connection

*Figure 8 continued on next page*

*Figure 8 continued*

probability in the follower population, $P_{PP}$. (**F**) Refractory period and delay from stimulation to burst as functions of the connection probability for the simulations shown in (**A–E**), still with $N_P \cdot P_{PP} \cdot W_{PP} = constant$. Error bars indicate SD. Notice that the refractory period increases with increasing connection probability. (**G**) Effect of $P_{PP}$ on the delay to evoked bursts. (**H**) Probability of evoking a burst as a function of time of stimulation delivery (colorbar) and the number out of nine stimulated neurons that are follower neurons for the baseline case of 2% connection probability.

The online version of this article includes the following source data for figure 8:

**Source data 1.** Refractory period of evoked bursts following holographic stimulation.

---

s refractory period, as indicated by the positive slope in *Figure 8H*. Additionally, eliciting a network burst does not require stimulation of rhythmogenic neurons.

## Discussion

Recent experiments have revealed a decoupling of respiratory rhythm generation and output patterning in the preBötC, which has given rise to the conceptual framework of burstlet theory. To date, however, this theory lacks the quantitative basis, grounded in underlying biophysical mechanisms, needed for its objective evaluation. To address this critical gap, in this computational study we developed a data-constrained biophysical model of the preBötC that generates burstlets and bursts as proposed by burstlet theory, with a range of features that match experimental observations. To summarize, we first show that CICR from intracellular stores is a natural mechanism to periodically amplify postsynaptic calcium transients needed for $I_{CAN}$ activation and recruitment of pattern-forming neurons into network bursts (*Figure 1*). Next, we demonstrate that in a two-neuron network CICR can convert baseline rhythmic activity into a mixture of bursts and burstlets, where the burstlet fraction depends largely on the magnitude of postsynaptic calcium transients (*Figure 2*). In a larger preBötC network containing rhythm- and pattern-forming subpopulations with experimentally constrained intrinsic properties, population sizes, and synaptic connectivity probabilities (*Figure 3*), similar but more realistic activity patterns arise (*Figure 4*). Moreover, we show that this model can match a wide range of the key experimental underpinnings of burstlet theory: dependence of the burstlet fraction on extracellular potassium concentration (*Figure 4I*), the Ca²⁺ dependence of the burstlet-to-burst transition (*Figure 5*), the effects of opioids on burst frequency and burstlet fraction (*Figure 6*), and the long delay and refractory period of bursts evoked by holographic photostimulation of small subsets of preBötC neurons (*Figures 7 and 8*).

### Insights into the mechanisms of burst (pattern) and burstlet (rhythm) generation within the inspiratory preBötC

Burstlet theory to date has largely been an empirical description of the observed features of bursts and burstlets. One idea that has been suggested is that rhythm generation is driven by a stochastic percolation process in which tonic spiking across the rhythm-generating population gradually synchronizes during the inter-burst-interval to generate the burstlet rhythm. Subsequently, a burst (i.e., motor output) only occurs if the burstlet is of sufficient magnitude, resulting from sufficient synchrony, to trigger all-or-none recruitment of the pattern-forming subpopulation (*Kam et al., 2013a*; *Kam et al., 2013b*; *Feldman and Kam, 2015*; *Cui et al., 2016*; *Kallurkar et al., 2020*; *Ashhad and Feldman, 2020*). This theory, however, does not identify or propose specific biophysical mechanisms capable of generating a quantitative explanation of the underlying cellular and network-level dynamics, fails to capture the Ca²⁺ dependence of the burst-to-burstlet transition, and cannot explain how extracellular potassium concentration impacts the burstlet fraction. Our simulations support an alternative view of the recruitment process associated with this transition that builds directly from previous computational studies (*Jasinski et al., 2013*; *Phillips et al., 2019a*; *Phillips and Rubin, 2019b*; *Phillips et al., 2021*), which robustly reproduce a wide array of experimental observations. Specifically, in this study we show that amplification of postsynaptic calcium transients in the pattern-generating subpopulation (triggered by burstlets) provides a natural mechanism capable of explaining the Ca²⁺ dependence of the burstlet-to-burst transition.

Importantly, our model yields the result, and hence the prediction, that the burstlet fraction is determined by the probability that a burstlet will trigger CICR in the pattern-forming subpopulation. In the

model, this probability is determined by the magnitude of postsynaptic calcium transients as well as the activation dynamics of the IP3 receptor and the SERCA pump. Therefore, to explain the decrease in the burstlet fraction with increasing extracellular $K_{bath}$, the magnitude of the burstlet-triggered postsynaptic calcium transients must increase with $K_{bath}$. Some of this rise can result directly from the increase in burstlet amplitude with increasing $K_{bath}$ (see *Kallurkar et al., 2020* and *Figure 4C*). To fully match the experimentally observed relationship between $K_{bath}$ and the burstlet fraction (*Figure 4J*), we also explicitly increased the parameter $P_{SynCa}$, which sets the proportion of the postsynaptic calcium current carried by $Ca^{2+}$. Thus, our model predicts that the magnitude of postsynaptic $Ca^{2+}$ transients triggered by EPSPs should increase as $K_{bath}$ is elevated.

This same prediction arises from considering the voltage-dependent properties of $Ca^{2+}$ channels characterized in preBötC neurons and the changes in the membrane potential of non-rhythmogenic (i.e., pattern-forming) neurons as a function of $K_{bath}$. Specifically, it is likely that voltage-gated calcium channels are involved in generating the postsynaptic $Ca^{2+}$ transients as dendritic $Ca^{2+}$ transients have been shown to precede inspiratory bursts and to be sensitive to $Cd^{2+}$, a calcium channel blocker (*Del Negro et al., 2011*). Consistent with this idea, $Cd^{2+}$-sensitive $Ca^{2+}$ channels in preBötC neurons appear to be primarily localized in distal dendritic compartments (*Phillips et al., 2018*). Voltage-gated calcium channels in the preBötC start to activate at approximately $-65\,\mathrm{mV}$ (*Elsen and Ramirez, 1998*), and importantly, the mean somatic resting membrane potential of non-rhythmogenic preBötC neurons increases from $-67.034\,\mathrm{mV}$ to $-61.78\,\mathrm{mV}$ when extracellular potassium concentration is elevated from $3\,\mathrm{mM}$ to $8\,\mathrm{mM}$ (*Tryba et al., 2003*). Moreover, at $K_{bath} = 9\,\mathrm{mM}$, EPSPs in the preBötC are on the order of 2–5 mV (*Kottick and Del Negro, 2015*; *Morgado-Valle et al., 2015*; *Baertsch et al., 2021*) and the amplitude of EPSCs has been shown to decrease as $K_{bath}$ is lowered (*Okada et al., 2005*). Putting together these data on resting membrane potential, EPSP sizes, and voltage-dependent activation of $Ca^{2+}$ channels, we deduce that when $K_{bath} = 3\,\mathrm{mM}$, the magnitude of EPSPs may not reach voltages sufficient for significant activation of voltage-gated $Ca^{2+}$ channels. As $K_{bath}$ is increased, however, both EPSC magnitudes and the membrane potential of pattern-forming neurons increase. Therefore, with increased $K_{bath}$, the prediction is that EPSCs will result in greater activation of voltage-gated $Ca^{2+}$ channels and increased postsynaptic calcium transients. This effect is captured in the model by an increase in the parameter $P_{SynCa}$, which determines the percentage of the postsynaptic current carried by $Ca^{2+}$ ions, with $K_{bath}$.

The idea that dendritic postsynaptic $Ca^{2+}$ transients and $I_{CAN}$ activation play a critical role in regulating the pattern of preBötC dynamics is well supported by experimental and computational studies. Specifically, the dendritic $Ca^{2+}$ transients that precede inspiratory bursts (*Del Negro et al., 2011*) have been shown to travel in a wave to the soma, where they activate TRPM4 currents ($I_{CAN}$) (*Mironov, 2008*). Moreover, the rhythmic depolarization of otherwise non-rhythmogenic neurons (inspiratory drive potential) depends on $I_{CAN}$ (*Pace et al., 2007a*), while the inspiratory drive potential is not dependent on $Ca^{2+}$ transients driven by voltage-gated calcium channels expressed in the soma (*Morgado-Valle et al., 2008*). Finally, pharmacological blockade of TRPM4 channels, thought to represent the molecular correlates of $I_{CAN}$, reduces the amplitude of preBötC motor output without impacting the rhythm (*Koizumi et al., 2018*; *Picardo et al., 2019*). These experimental findings were incorporated into and robustly reproduced in a recent computational model (*Phillips et al., 2019a*) and are reproduced here (see *Figure 5B and D*). Consistent with these findings, this previous model suggests that rhythm generation arises from a small subset of preBötC neurons, which form an $I_{NaP}$-dependent rhythmogenic kernel (i.e., burstlet rhythm generator), and that rhythmic synaptic drive from these neurons triggers postsynaptic calcium transients, $I_{CAN}$ activation, and amplification of the inspiratory drive potential, which spurs bursting in the rest of the network. This study builds on this previous model by explicitly defining rhythm- and pattern-generating neuronal subpopulations (see *Figure 3*) and incorporating the mechanisms required for CICR and intermittent amplification of postsynaptic calcium transients.

CICR mediated by the SERCA pump and the IP3 receptor has long been suspected to be involved in the dynamics of preBötC rhythm and/or pattern generation (*Pace et al., 2007a*; *Crowder et al., 2007*; *Mironov, 2008*; *Toporikova et al., 2015*) and has been explored in individual neurons and network models of the preBötC (*Toporikova and Butera, 2011*; *Jasinski et al., 2013*; *Rubin et al., 2009*; *Wang and Rubin, 2020*). Experimental studies have not clearly established the role of CICR from ER stores in respiratory circuits, however. For example, *Mironov, 2008* showed that application

of 1 μM thapsigargin, a SERCA pump inhibitor, abolished rhythmic activity and blocked calcium transients that travel in a wave from the dendrites to the soma. In a separate study, however, block of the SERCA pump by bath application of thapsigargin (2–20 μM) or cyclopiazonic acid (CPA) (30–50 μM) did not significantly affect the amplitude or frequency of hypoglossal motor output in in vitro slice preparations containing the preBötC (*Beltran-Parrazal et al., 2012*). The explanation for these seemingly contradictory experimental results is unclear, especially since effects of SERCA pump block could be complicated, and will need to be investigated by future studies. It is possible that the role of CICR may be dynamically regulated depending on the state of the preBötC network. For example, the calcium concentration at which the IP3 receptor is activated is dynamically regulated by IP3 (*Kaftan et al., 1997*), and therefore, activity- or neuromodulatory-dependent changes in the cytoplasmic $Ca^{2+}$ and/or IP3 concentration may impact ER $Ca^{2+}$ uptake and release dynamics. Store-operated $Ca^{2+}$ dynamics are also affected by the transient receptor potential canonical 3 (TRPC3) channels (*Salido et al., 2009*), which are expressed in the preBötC, and manipulation of TRPC3 has been shown to impact burst amplitude and regularity (*Tryba et al., 2003*; *Koizumi et al., 2018*) as would be predicted by this model. It is also possible that calcium release is mediated by the ryanodine receptor, an additional calcium-activated channel located in the ER membrane (*Lanner et al., 2010*), since bath application of CPA (100 μM) and ryanodine (10 μM) removed large-amplitude oscillations in recordings of preBötC population activity (*Toporikova et al., 2015*).

Finally, we note that while various markers can be used to define distinct subpopulations of neurons within the preBötC, our model cannot determine which of these ensembles are responsible for rhythm and pattern generation. Past experiments have examined the impact of optogenetic inhibition, applied at various intensities to subpopulations associated with specific markers, on the frequency of inspiratory neural activity, but this activity was measured by motor output, not within the preBötC itself (*Tan et al., 2008*; *Cui et al., 2016*; *Koizumi et al., 2016*). According to burstlet theory and our model, slowed output rhythmicity could derive from inhibition of rhythm-generating neurons, due to a reduced frequency of burstlets, and from inhibition of pattern-generating neurons, due to a reduced success rate of burst recruitment. Thus, measurements within the preBötC will be needed in order to assess the mapping between subpopulations of preBötC neurons and roles in burstlet and burst production.

## Additional comparisons to experimental results

In our model (*Figure 4*), a burstlet rhythm first emerges at a $K_{bath}$ of approximately 5 mM, whereas in the experiments of *Kallurkar et al., 2020*, the burstlet rhythm continues even down to 3 mM. To explain this discrepancy, we note that our model assumes that the extracellular potassium concentration throughout the network is equal to $K_{bath}$. Respiratory circuits appear to have some buffering capacity, however, such that for $K_{bath}$ concentrations below approximately 5 mM the extracellular K⁺ concentration remains elevated above $K_{bath}$ (*Okada et al., 2005*). The $K_{bath}$ range over which our model generates a rhythm would extend to that seen experimentally if extracellular K⁺ buffering were accounted for. This buffering effect can also explain why the burstlet fraction remains constant in experimental studies when $K_{bath}$ is lowered from 5 mM to 3 mM (*Kallurkar et al., 2020*). Our model also does not incorporate short-term extracellular potassium dynamics that depend on $K_{bath}$ and may impact the ramping shape of burstlet onset (*Abdulla et al., 2021*). Importantly, over the range of $K_{bath}$ values relevant both to experiments and our model, we find clear agreement on the dependence of burstlet fraction on $K_{bath}$ (*Figure 4I*).

Although our model incorporates various key biological features, it does not include some of the biophysical mechanisms that are known to shape preBötC patterned output or that are hypothesized to contribute to the properties described by burstlet theory. For example, the M-current associated with KCNQ potassium channels has been shown to impact burst duration by contributing to burst termination (*Revill et al., 2021*). Additionally, intrinsic conductances associated with a hyperpolarization-activated mixed cation current ($I_h$) and a transient potassium current ($I_A$) are hypothesized to be selectively expressed in the pattern- and rhythm-generating preBötC subpopulations (*Picardo et al., 2013*; *Phillips et al., 2018*). Thus, our model predicts that while these currents may impact quantitative properties of burstlets and bursts, they are not critical for the presence of burstlets and their transformation into bursts. The current model also does not include a population of inhibitory preBötC neurons. Inhibition is involved in regulating burst amplitude (*Baertsch et al., 2018*), but it does not

have a clear role in burst or burstlet generation, and therefore inhibition was omitted from this work. More globally, it is crucial to recognize that areas outside of the preBötC impact dynamics within the preBötC. These effects, which remain to be fully elucidated, may range from ongoing modulation of the level of excitability of preBötC neurons to timed signaling that contributes to preBötC rhythmicity and patterning (e.g., *Mulkey et al., 2004*; *Dutschmann and Dick, 2012*; *Phillips et al., 2012*; *Smith et al., 2013*; *Dhingra et al., 2019*; *Richter et al., 2019*; *Liu et al., 2022*). For example, transection studies suggest that pontine regions may make crucial contributions to respiratory circuit excitability and respiratory pattern formation (*Jones and Dutschmann, 2016*; *Smith et al., 2007*). Finally, the data on which this study was based comes from a variety of settings, including in vitro and other reduced preparations, and additional factors no doubt complicate the generation and control of respiratory outputs in vivo. Indeed, although experimental results suggest that manipulations to enhance preBötC excitability in slice preparations do not appear to significantly impact the mechanisms of preBötC rhythmicity or the generation of bursts and burstlets, additional investigation of how higher brainstem centers impact preBötC inspiratory rhythm and pattern generation is an important direction for future studies.

Importantly, our model does robustly reproduce all of the range of key experimental observations underlying burstlet theory. Not surprisingly, block of calcium transients or $I_{CAN}$ in our model eliminates bursts without affecting the underlying rhythm (*Figure 5*), which is consistent with experimental observations (*Kam et al., 2013b*; *Sun et al., 2019*). Interestingly, our model also provides the experimentally testable predictions that blocking calcium transients will increase the burstlet fraction while $I_{CAN}$ block will have no effect on this fraction, whereas both perturbations will smoothly reduce burst amplitude. The calcium-dependent mechanisms that we include in our model pattern-generating population have some common features with a previous model that suggested the possible existence of two distinct preBötC neuronal populations responsible for eupneic burst and sigh generation, respectively, which also included excitatory synaptic transmission from the former to the latter (*Toporikova et al., 2015*). In the eupnea-sigh model, however, the population responsible for low-frequency, high-amplitude sighs was capable of rhythmic burst generation even without synaptic drive, in contrast to the pattern-generation population as tuned in our model. Also, in contrast to the results on bursts considered in our study, sigh frequency in the earlier model did not vary with extracellular potassium concentration and sigh generation required a hyperpolarization-activated inward current, $I_h$.

We also considered the effects of opioids in the context of burstlets and bursts, a topic that has not been extensively studied. It is well established that opioids slow the preBötC rhythm in in vitro slice preparations; however, the limited results presented to date on effects of opioids on the burstlet fraction are inconsistent. Specifically, *Sun et al., 2019* found that application of the μOR agonist DAMGO at 10 nM and 30 nM progressively decreased the preBötC network frequency but had no impact on the burstlet fraction before the network rhythm was eventually abolished at approximately 100 nM. Similarly, *Baertsch et al., 2021* found that DAMGO decreased the preBötC network frequency in a dose-dependent fashion; however, in these experiments the network was less sensitive to DAMGO, maintaining rhythmicity up to approximately 300 nM, and the burstlet fraction increased with increasing DAMGO concentration. The inconsistent effects of DAMGO on the burstlet fraction across these two studies can be explained by our simulations based on the different sensitivities of these two preparations to DAMGO and the two distinct mechanisms of action of DAMGO on neurons that express μOR – decreases in excitability and decreases in synaptic output of neurons – identified by *Baertsch et al., 2021*. In our simulations, we show that the decreased excitability resulting from activation of a GIRK channel only impacts frequency, whereas decreasing the synaptic output of μOR-expressing neurons results in an increase in the burstlet fraction and a decrease in burst frequency (*Figure 6*). In experiments, suppression of synaptic output does not appear to occur until DAMGO concentrations are above approximately 50 nM(*Baertsch et al., 2021*). Therefore, it is not surprising that DAMGO application did not strongly impact the burstlet fraction before the rhythm was ultimately abolished in *Sun et al., 2019* due to the higher DAMGO sensitivity of that particular experimental preparation, as indicated by the lower dose needed for rhythm cessation.

## Mixed-mode oscillations

Mixed-mode oscillations, in which intrinsic dynamics of a nonlinear system naturally lead to alternations between small- and large-amplitude oscillations (*Del Negro et al., 2002c*; *Bertram and Rubin,*

*2017*), are a mechanism that has been previously proposed to underlie bursts and burstlets under the assumption of differences in intrinsic oscillation frequencies across preBötC neurons (*Bacak et al., 2016*). This mechanism was not needed to explain the generation of bursts and burstlets in the current model, however. Moreover, systems with mixed-mode oscillations can show a wide range of oscillation amplitudes under small changes in conditions and mixed-mode oscillations only emerge in the preBötC when $K_{bath}$ is elevated above 9 mM (*Del Negro et al., 2002c*). These properties are not consistent with the burst and burstlet amplitudes or $K_{bath}$-dependent changes in the burstlet fraction seen experimentally (*Kallurkar et al., 2020*) and in our model.

## Holographic photostimulation, percolation, and rhythm generation

Experimental data supporting burstlet theory has shown that burstlets are the rhythmogenic event in the preBötC. However, although burstlet theory is sometimes referenced as a theory of respiratory rhythm generation, the actual mechanisms of burstlet rhythm generation remain unclear. One idea that has been suggested is that rhythm generation is driven by a stochastic percolation process in which tonic spiking across the rhythm-generating population gradually synchronizes during the inter-burst-interval to generate the burstlet rhythm (*Ashhad and Feldman, 2020*; *Slepukhin et al., 2020*). In this framework, a burst (i.e., motor output) only occurs if the burstlet is of sufficient magnitude, resulting from sufficient synchrony, to trigger all-or-none recruitment of the pattern-forming subpopulation (*Kam et al., 2013a*; *Kam et al., 2013b*; *Feldman and Kam, 2015*; *Kallurkar et al., 2020*; *Ashhad and Feldman, 2020*; *Slepukhin et al., 2020*).

The idea that burstlets are the rhythmogenic event within the preBötC is supported by the observation that block of voltage-gated $Ca^{2+}$ channels by $Cd^{2+}$ eliminates bursts without affecting the underlying burstlet rhythm (*Kam et al., 2013a*; *Sun et al., 2019*). However, the rhythmogenic mechanism based on percolation is speculative and comes from two experimental observations. The first is that the duration and slope (i.e., shape) of the burstlet onset are statistically indistinguishable from the ramping pre-inspiratory activity that immediately precedes inspiratory bursts (*Kallurkar et al., 2020*). We note, however, that this shape of pre-inspiratory activity can arise through intrinsic mechanisms at the individual neuron level (*Abdulla et al., 2021*). The second observation evoked in support of the percolation idea is that holographic photostimulation of small subsets (4–9) of preBötC neurons can elicit bursts with delays lasting hundreds of milliseconds (*Kam et al., 2013b*). These delays are longer than could be explained with existing preBötC models and have approximately the same duration as the pre-inspiratory activity and burstlet onset hypothesized to underlie the rhythm. According to the percolation hypothesis of burstlet rhythm generation, these long delays result from the specific topological architecture of the preBötC, recently proposed to be a heavy-tailed synaptic weight distribution in the rhythmogenic preBötC subpopulation (*Slepukhin et al., 2020*).

Interestingly, the model presented here naturally captures the long delays characterized by *Kam et al., 2013b*, and stimulation of small subsets of neurons triggers a growth in population activity in the lead up to a burst that could be described as percolation (*Figure 7B*). Our model does not require a special synaptic weight distribution to generate the long delays, however. Indeed, our model suggests that the long delays between simulation and burst generation are due in large part to the dynamics of the pattern-forming population, as probabilistically these neurons are the most likely to be stimulated and they appear to play a dominant role in setting the timing of the elicited burst response (*Figure 8H*). Moreover, the dynamics of this population is strongly impacted by the CICR mechanism proposed here, which is required for burst generation. Interestingly, to match the 500 ms refractory period following an endogenous burst during which holographic stimulation cannot elicit a burst, our model predicts that the connection probability in the pattern-generating preBötC subpopulation must be between 1% and 2% (*Figure 8A and B*), which is consistent with available experimental data (*Ashhad and Feldman, 2020*). Experiments applying global, presumably weaker stimulation to the preBötC yield longer (~2 s) refractory periods after endogenous bursts (*Baertsch et al., 2018*; *Kottick and Del Negro, 2015*), and our model can also produce similar refractory periods in analogous conditions.

Thus, taken together, previous modeling and our work offer two alternative, seemingly viable hypotheses about the source of the delay between holographic stimulation and burst onset, each related to a proposed mechanism for burstlet and burst generation. Yet additional arguments call into question aspects of the percolation idea. If the burstlet rhythm is driven by a stochastic percolation

process, then the period and amplitude of burstlets should be stochastic, irregular, and broadly distributed. Moreover, in the proposed framework of burstlet theory, the pattern of bursts and burstlets for a given burstlet fraction would also be stochastic since the burstlet-to-burst transition is thought to be an all-or-none process that depends on the generation of a burstlet of sufficient magnitude. Example traces illustrating a mixture of bursts and burstlets typically show a pattern of multiple burstlets followed by a burst that appears to regularly repeat (*Kam et al., 2013b*; *Sun et al., 2019*; *Kallurkar et al., 2020*) and hypoglossal output timing has also been found to exhibit high regularity *Kam et al., 2013b*, however, suggesting that the burstlet-to-burst transition is not dependent on the synchrony and hence magnitudes of individual burstlets but rather on a slow process that gradually evolves over multiple burstlets. The regularity and patterns of burstlets and bursts that arise from such a process in our model match well with those observed experimentally.

We note that the burstlet-to-burst transition mechanism proposed here, based on CICR from ER stores, depends on rhythmic inputs from the rhythm-generating to the pattern-generation population; however, it is independent of the mechanism of rhythm generation. In our simulations, rhythm generation depends on the slowly inactivating persistent sodium current ($I_{NaP}$). The role of $I_{NaP}$ in preBötC inspiratory rhythm generation is a contentious topic within the field, largely due to the inconsistent effects of $I_{NaP}$ block. We chose to use $I_{NaP}$ as the rhythmogenic mechanism in the burstlet population for a number of reasons: (1) consideration of the pharmacological mechanism of action and nonuniform effects of drug penetration can explain the seemingly contradictory experimental findings relating to $I_{NaP}$ (*Phillips and Rubin, 2019b*), (2) $I_{NaP}$-dependent rhythm generation is a well-established and understood idea (*Butera et al., 1999*), (3) recent computational work on which the current model is based suggests that rhythm generation occurs in a small, $I_{NaP}$-dependent rhythmogenic kernel that is analogous to the burstlet population (*Phillips et al., 2019a*), and predictions from this model that depend on the specific proposed roles of $I_{NaP}$ and $I_{CAN}$ in rhythm and pattern formation have been experimentally confirmed in a recent study (*Phillips et al., 2021*). It is important to note, however, that the findings about burstlets and bursts presented in this work would have been obtained if the burstlet rhythm was imposed (*Figure 4—figure supplement 1*) or if burstlets were generated by some other means, such as by the percolation mechanism proposed by burstlet theory.

## Summary of model predictions

The model presented here is itself a prediction; that is, this work predicts that a CICR-mediated mechanism is critical to the transition of burstlets into bursts. At a more specific level, our model makes the following predictions: (1) the magnitude of postsynaptic calcium transients triggered by EPSCs in preBötC neurons will increase with K$^+$ (see *Figure 4* and related text); (2) network-level burstlets and bursts will persist if currents involved in regulating burst shape, such as $I_h$ and $I_A$, are blocked (see earlier discussion); (3) blocking postsynaptic Ca$^{2+}$ transients will increase the burstlet fraction and decrease the burst amplitude before network bursts are eventually abolished (see *Figure 5*); (4) $I_{CAN}$ block will not change the burstlet fraction and will decrease burstlet amplitudes (see *Figure 5*); (5) the synaptic connection probability within the pattern-generating population in the preBötC is low (1–2%, see *Figure 8*); and (6) selective holographic stimulation of pattern-generating neurons should be more effective than stimulation of rhythm-generating neurons at triggering network bursts (see *Figure 8*). This could be tested by selectively stimulating Dbx1 preBötC neurons that express Sst (pattern forming) or that do not express Sst (rhythmogenic).

## Conclusions

This study has developed the first model-based description of the biophysical mechanism underlying the generation of bursts and burstlets in the inspiratory preBötC. As suggested by burstlet theory and other previous studies, rhythm and pattern generation in this work are represented by two distinct preBötC subpopulations. A key feature of our model is that generation of network bursts (i.e., motor output) requires amplification of postsynaptic Ca$^{2+}$ transients by CICR in order to activate $I_{CAN}$ and drive bursting in the rest of the network. Moreover, the burstlet fraction depends on rate of Ca$^{2+}$ buildup in intracellular stores, which is impacted by $K_{bath}$-dependent modulation of preBötC excitability. These ideas complement other recent findings on preBötC rhythm generation (*Phillips et al., 2019a*; *Phillips and Rubin, 2019b*; *Phillips et al., 2021*), together offering a unified explanation for a

large body of experimental findings on preBötC inspiratory activity that form a theoretical foundation on which future developments can build.

# Materials and methods

## Neuron model

Model preBötC neurons include a single compartment and incorporate Hodgkin–Huxley-style conductances adapted from previously described models (*Jasinski et al., 2013*; *Phillips et al., 2019a*; *Phillips and Rubin, 2019b*) and/or experimental data as detailed below. The membrane potential of each neuron is governed by the following differential equation:

$$C\frac{dV}{dt} = -I_{Na} - I_K - I_{NaP} - I_{Ca} - I_{CAN} - I_{Leak} - I_{Syn} - I_{GIRK} - I_{Holo} + I_{APP}, \tag{1}$$

where $C = 36\,\text{pF}$ is the membrane capacitance and each $I_i$ represents a current, with $i$ denoting the current's type. The currents include the action potential generating Na$^+$ and delayed rectifying K$^+$ currents ($I_{Na}$ and $I_K$), persistent Na$^+$ current ($I_{NaP}$), voltage-gated Ca$^{2+}$ current ($I_{Ca}$), Ca$^{2+}$-activated nonselective cation (CAN) current ($I_{CAN}$), K$^+$-dominated leak current ($I_{Leak}$), synaptic current ($I_{Syn}$), μ-opioid receptor-activated G protein-coupled inwardly rectifying K$^+$ leak current ($I_{GIRK}$) (*Kubo et al., 1993*), and a holographic photostimulation current ($I_{Holo}$). $I_{APP}$ denotes an applied current injected from an electrode. The currents are defined as follows:

$$I_{Na} = g_{Na} \cdot m_{Na}^3 \cdot h_{Na} \cdot (V - E_{Na}) \tag{2}$$

$$I_K = g_K \cdot m_K^4 \cdot (V - E_K) \tag{3}$$

**Table 1.** Ionic channel parameters.

| Channel | Parameters | | | | |
|---|---|---|---|---|---|
| $I_{Na}$ | $g_{Na} = 150\,\text{nS}$ | $E_{Na} = 26.54 \cdot ln(Na_{out}/Na_{in})$ | $Na_{in} = 15\,\text{mM}$ | $Na_{out} = 120\,\text{mM}$ | |
| | $m_{1/2} = -43.8\,\text{mV}$ | $k_m = 6.0\,\text{mV}$ | $\tau_{max}^m = 0.25\,\text{ms}$ | $\tau_{1/2}^m = -43.8\,\text{mV}$ | $k_\tau^m = 14.0\,\text{mV}$ |
| | $h_{1/2} = -67.5\,\text{mV}$ | $k_h = -11.8\,\text{mV}$ | $\tau_{max}^h = 8.46\,\text{ms}$ | $\tau_{1/2}^h = -67.5\,\text{mV}$ | $k_\tau^h = 12.8\,\text{mV}$ |
| $I_K$ | $g_K = 220\,\text{nS}$ | $E_K = 26.54 \cdot ln(K_{bath}/K_{in})$ | $K_{in} = 125$ | $K_{Bath} = VAR$ | |
| | $A_\alpha = 0.011$ | $B_\alpha = 44.0\,\text{mV}$ | $k_\alpha = 5.0\,\text{mV}$ | | |
| | $A_\beta = 0.17$ | $B_\beta = 49.0\,\text{mV}$ | $k_\beta = 40.0\,\text{mV}$ | | |
| $I_{NaP}$ | $g_{NaP} = N(\mu, \sigma)$, see *Table 2* | | | | |
| | $m_{1/2} = -47.1\,\text{mV}$ | $k_m = 3.1\,\text{mV}$ | $\tau_{max}^m = 1.0\,\text{ms}$ | $\tau_{1/2}^m = -47.1\,\text{mV}$ | $k_\tau^m = 6.2\,\text{mV}$ |
| | $h_{1/2} = -60.0\,\text{mV}$ | $k_h = -9.0\,\text{mV}$ | $\tau_{max}^h = 5000\,\text{ms}$ | $\tau_{1/2}^h = -60.0\,\text{mV}$ | $k_\tau^h = 9.0\,\text{mV}$ |
| $I_{Ca}$ | $g_{Ca} = 0.0065\,\text{pS}$ | $E_{Ca} = 13.27 \cdot ln(Ca_{out}/Ca_{in})$ | | $Ca_{out} = 4.0\,\text{mM}$ | |
| | $m_{1/2} = -27.5\,\text{mV}$ | $k_m = 5.7\,\text{mV}$ | $\tau_m = 0.5\,\text{ms}$ | | |
| | $h_{1/2} = -52.4\,\text{mV}$ | $k_h = -5.2\,\text{mV}$ | $\tau_h = 18.0\,\text{ms}$ | | |
| $I_{CAN}$ | $g_{CAN} = N(\mu, \sigma)$, see *Table 2* | $E_{CAN} = 0.0\,\text{mV}$ | $Ca_{1/2} = 0.00074\,\text{mM}$ | $n = 0.97$ | |
| $I_{Leak}$ | $g_{Leak} = N(\mu, \sigma)$, see *Table 2* | $E_{Leak} = -26.54 * ln[(P_{Na} * Na_{in} + P_K * K_{in})/(P_{Na} * Na_{out} + P_K * K_{bath})]$ | | | |
| | $P_{Na} = 1$ | $P_K = 42$ | | | |
| $I_{Syn}$ | $g_{Syn} = VAR$, see *Equation 25* | $E_{Syn} = 0.0\,\text{mV}$ | $\tau_{Syn} = 5.0\,\text{ms}$ | | |
| $I_{GIRK}$ | $g_{GIRK} = 0 - 0.3\,\text{nS}$ | $E_{GIRK} = E_K$ | | | |
| $I_{Holo}$ | $g_{Holo} = 50\,\text{nS}$ | $\tau_{Holo} = 100\,\text{ms}$ | $E_{Holo} = E_{Syn}$ | | |

**Table 2.** Distributed channel conductances.

| Type | $g_{NaP}$ (nS) | | $g_{Leak}$ (nS) | | $g_{CAN}$ (nS) | |
|---|---|---|---|---|---|---|
| | μ | σ | μ | σ | μ | σ |
| Rhythm | 3.33 | 0.75 | $exp((K_{Bath} - 3.425)/4.05)$ | $0.05 \cdot \mu_{leak}$ | 0.0 | 0.0 |
| Pattern | 1.5 | 0.25 | $exp((K_{Bath} - 3.425)/4.05)$ | $0.025 \cdot \mu_{leak}$ | 2.0 | 1.0 |

$$I_{NaP} = g_{NaP} \cdot m_{NaP} \cdot h_{NaP} \cdot (V - E_{Na}) \tag{4}$$

$$I_{Ca} = g_{Ca} \cdot m_{Ca} \cdot h_{Ca} \cdot (V - E_{Ca}) \tag{5}$$

$$I_{CAN} = g_{CAN} \cdot m_{CAN} \cdot (V - E_{CAN}) \tag{6}$$

$$I_{Leak} = g_{Leak} \cdot (V - E_{Leak}) \tag{7}$$

$$I_{Syn} = g_{Syn} \cdot (V - E_{Syn}) \tag{8}$$

$$I_{GIRK} = g_{GIRK} \cdot (V - E_K) \tag{9}$$

$$I_{Holo} = g_{Holo} \cdot (V - E_{Holo}) \tag{10}$$

where $g_i$ is the maximum conductance, $E_i$ is the reversal potential, and $m_i$ and $h_i$ are gating variables for channel activation and inactivation for each current $I_i$. The glutamatergic synaptic conductance $g_{Syn}$ is dynamic and is defined below. The values used for the $g_i$ and $E_i$ are mostly shown in **Table 1**, with a few conductances selected from distributions as indicated in **Table 2**.

Activation ($m_i$) and inactivation ($h_i$) of voltage-dependent channels are described by the following differential equation:

$$\tau_X(V) \cdot \frac{dX}{dt} = X_\infty(V) - X; \quad X \in \{m, h\} \tag{11}$$

where $X_\infty$ represents steady-state activation/inactivation and $\tau_X$ is a time constant. For $I_{Na}$, $I_{NaP}$, and $I_{Ca}$, the functions $X_\infty$ and $\tau_X$ take the forms

$$X_\infty(V) = 1/(1 + \exp(-(V - X_{1/2})/k_X)), \tag{12}$$

$$\tau_X(V) = \tau_{max}^X / \cosh((V - \tau_{1/2}^X)/k_\tau^X). \tag{13}$$

The values of the parameters ($X_{1/2}$, $k_X$, $\tau_{max}^X$, $\tau_{1/2}^X$, and $k_\tau^X$) corresponding to $I_{Na}$, $I_{NaP}$ and $I_{Ca}$ are given in **Table 1**.

For the voltage-gated potassium channel, the steady-state activation $m_\infty^K(V)$ and time constant $\tau_m^K(V)$ are given by the expressions

$$m_\infty^K(V) = \alpha_\infty(V)/(\alpha_\infty(V) + \beta_\infty(V)), \tag{14}$$

$$\tau_m^K(V) = 1/(\alpha_\infty(V) + \beta_\infty(V)) \tag{15}$$

where

$$\alpha_\infty(V) = A_\alpha \cdot (V + B_\alpha)/(1 - \exp(-(V + B_\alpha)/k_\alpha)), \tag{16}$$

$$\beta_\infty(V) = A_\beta \cdot \exp(-(V + B_\beta)/k_\beta). \tag{17}$$

The values for the constants $A_\alpha$, $A_\beta$, $B_\alpha$, $B_\beta$, $k_\alpha$, and $k_\beta$ are also given in **Table 1**.

$I_{CAN}$ activation depends on the $Ca^{2+}$ concentration in the cytoplasm ($[Ca]_{Cyto}$) and is given by

$$m_{CAN} = 1/(1 + (Ca_{1/2}/[Ca]_{Cyto})^n). \tag{18}$$

The parameters $Ca_{1/2}$ and $n$ represent the half-activation $Ca^{2+}$ concentration and the Hill coefficient, respectively, and are included in **Table 1**.

The dynamics of $[Ca]_{Cyto}$ is determined in part by the balance of $Ca^{2+}$ efflux toward a baseline concentration via the $Ca^{2+}$ pump and $Ca^{2+}$ influx through voltage-dependent activation of $I_{Ca}$ and synaptically triggered $Ca^{2+}$ transients, with a percentage ($P_{SynCa}$) of the synaptic current ($I_{Syn}$) carried by $Ca^{2+}$ ions. Additionally, the model includes an intracellular compartment that represents the ER,

which impacts $[Ca]_{Cyto}$. The ER removes $Ca^{2+}$ from the cytoplasm via a sarcoplasmic/endoplasmic reticulum $Ca^{2+}$- ATPase (SERCA) pump, which transports $Ca^{2+}$ from the cytoplasm into the ER ($J_{SERCA}$) and releases $Ca^{2+}$ into the cytoplasm via calcium-dependent activation of the inositol triphosphate (IP3) receptor ($J_{IP3}$). Therefore, the dynamics of $[Ca]_{Cyto}$ is described by the following differential equation:

$$\frac{d[Ca]_{Cyto}}{dt} = \alpha_{Ca} \cdot (I_{Ca} + P_{SynCa} \cdot I_{Syn}) + \alpha_{ER} \cdot (J_{IP3} - J_{SERCA}) - \frac{([Ca]_{Cyto} - Ca_{min})}{\tau_{pump}}, \tag{19}$$

where $\alpha_{Ca} = 2.5 \cdot 10^{-5}$ mM/fC is a conversion factor relating current to rate of change of $[Ca]_{Cyto}$, $\tau_{pump} = 500$ ms is the time constant for the $Ca^{2+}$ pump, $Ca_{min} = 5.0 \cdot 10^{-6}$ mM is a minimal baseline calcium concentration, and $\alpha_{ER} = 2.5 \cdot 10^{-5}$ is the ratio of free to bound $Ca^{2+}$ in the ER.

The flux of $Ca^{2+}$ from the ER to the cytoplasm through the IP3 receptor is modeled as

$$J_{IP3} = \left( ER_{leak} + G_{IP3} \cdot \left( \frac{[Ca]_{Cyto}}{[Ca]_{Cyto} + K_a} \cdot \frac{[IP3]_i \cdot l}{[IP3]_i + K_l} \right)^3 \right) \cdot ([Ca]_{ER} - [Ca]_{Cyto}), \tag{20}$$

where $ER_{leak} = 0.1$/ms represents the leak constant from the ER stores, $G_{IP3} = 77,500$/ms represents the permeability of the IP3 channel, $K_a = 1.0 \cdot 10^{-4}$ mM and $K_l = 1.0 \cdot 10^{-3}$ mM are dissociation constants, and $[IP3]_i = 1.5 \cdot 10^{-3}$ mM is the cytoplasm IP3 concentration. Finally, the $Ca^{2+}$-dependent IP3 gating variable, $l$, and the $Ca^{2+}$ concentration in the ER, $[Ca]_{ER}$, are determined by the following equations:

$$\frac{dl}{dt} = A \cdot (K_d - l \cdot ([Ca]_{Cyto} + K_d)); \tag{21}$$

$$[Ca]_{ER} = ([Ca]_{total} - [Ca]_{Cyto})/\sigma_{Ca}, \tag{22}$$

where $A = 0.1$ mM/ms is a conversion factor, $K_d = 0.2 \cdot 10^{-3}$ mM is the dissociation constant for IP3 inactivation, $[Ca]_{total}$ is the total intracellular calcium concentration, and $\sigma_{Ca} = 0.185$ is the ratio of cytosolic to ER volume. The total intracellular calcium concentration is described as

$$\frac{d[Ca]_{Total}}{dt} = \alpha_{Ca} \cdot (I_{Ca} + P_{SynCa} \cdot I_{Syn}) - \frac{(Ca_{Cyto} - Ca_{min})}{\tau_{pump}}. \tag{23}$$

Finally, removal of $Ca^{2+}$ from the cytoplasm by the SERCA pump is modeled as

$$J_{SERCA} = G_{SERCA} \cdot \frac{[Ca]_{Cyto}^2}{K_{SERCA}^2 + [Ca]_{Cyto}^2}, \tag{24}$$

where $G_{SERCA} = 0.45$ mM/ms is the maximal flux through the SERCA pump, and $K_{SERCA} = 7.5 \cdot 10^{-5}$ mM is a dissociation constant.

Nondimensionalization of similar models in past work (**Wang and Rubin, 2017**; **Wang and Rubin, 2020**) has shown that $h_{NaP}, l$, and $[Ca]_{ER}$ are the slowest variables in the model and evolve on similar timescales, while $[Ca]_{Cyto}$ evolves on a faster timescale that is still significantly slower than that of the voltage dynamics and other current gating variables. Some subtleties arise in that different components of the calcium dynamics evolve on different timescales and their influences depend on the levels of calcium present in various domains within the cell, but these subtleties are not considered in this article.

When we include multiple neurons in the network, we can index them with subscripts. The total synaptic conductance $(g_{Syn})_i$ of the $i$th target neuron is described by the following equation:

$$(g_{Syn})_i = g_{Tonic} + \sum_{j \neq i; n} W_{j,i} \cdot D_j \cdot C_{j,i} \cdot H(t - t_{j,n}) \cdot e^{-(t - t_{j,n})/\tau_{syn}}, \tag{25}$$

where $g_{Tonic}$ is a fixed or tonic excitatory synaptic conductance (e.g., from respiratory control areas outside of the preBötC) that we assume impinges on all neurons, $W_{j,i}$ represents the weight of the synaptic connection from neuron $j$ to neuron $i$, $D_j$ is a scaling factor for short-term synaptic depression in the presynaptic neuron $j$ (described in more detail below), $C_{j,i}$ is an element of the connectivity matrix ($C_{j,i} = 1$ if neuron $j$ makes a synapse with neuron $i$ and $C_{j,i} = 0$ otherwise), $H(.)$ is the Heaviside step function, and $t$ denotes time. $\tau_{Syn}$ is an exponential synaptic decay constant, while $t_{j,n}$ is the time at which the $n$th action potential generated by neuron $j$ reaches neuron $i$.

We included synaptic depression in our model because experiments have revealed that it contributes to termination of inspiratory activity in the preBötC (**Kottick and Del Negro, 2015**) and past

**Table 3.** Maximal synaptic weights and connection probabilities between and within rhythm- and pattern-generating preBötC subpopulations ($W_{Max}, P$).

| | | Target | |
|---|---|---|---|
| | | Rhythm | Pattern |
| Source | Rhythm | (0.15 nS, 0.13) | (0.000175 nS, 0.3) |
| | Pattern | (0.25 nS, 0.3) | (0.0063 nS, 0.02) |

computational models have suggested that it might play an important role in preBötC network oscillations (*Rubin et al., 2009*; *Guerrier et al., 2015*). Synaptic depression in the $j$th neuron ($D_j$) was simulated using an established mean-field model of short-term synaptic dynamics (*Abbott et al., 1997*; *Dayan and Abbott, 2001*; *Morrison et al., 2008*) as follows:

$$\frac{dD_j}{dt} = \frac{D_0 - D_j}{\tau_D} - \alpha_D \cdot D_j \cdot \delta(t - t_j) \tag{26}$$

where the parameter $D_0 = 1$ sets the maximum value of $D_j$, $\tau_D = 1000$ ms sets the rate of recovery from synaptic depression, $\alpha_D = 0.2$ sets the fractional depression of the synapse each time neuron $j$ spikes, and $\delta(.)$ is the Kronecker delta function that equals 1 at the time of each spike in neuron $j$ and 0 otherwise. Parameters were chosen to qualitatively match data from *Kottick and Del Negro, 2015*. Note that with this choice of $\tau_D$ synaptic depression recovers on a timescale comparable to that of the other slowest variables in the model.

When we consider a two-neuron network (*Figure 2*), we take $W_{1,2} = W_{2,1} = 0.006$ and $C_{1,2} = C_{2,1} = 1$. For the full preBötC population model comprising rhythm- and pattern-generating subpopulations, the weights of excitatory conductances were uniformly distributed such that $W_{j,i} = U(0, W_{Max})$ where $W_{Max}$ is a constant associated with the source and target neurons' populations; with each such pair, we also associated a connection probability and used this to randomly set the $C_{j,i}$ values (see *Table 3*). Effects of opioids on synaptic transmission for source neurons in the rhythmogenic subpopulation (*Figure 6*) were simulated by scaling $W_{j,i}$ with the parameter $\gamma_{\mu OR}$, which ranged between 0 and 0.5 and sets the percent synaptic block.

## Network construction

The relative proportions of neurons assigned to the rhythm- and pattern-generating preBötC subpopulations were chosen based on experimental data. For example, *Kallurkar et al., 2020* found that $20 \pm 9\%$ of preBötC inspiratory neurons are active during burstlets at $K_{Bath} = 9$ mM. Moreover, the rhythm- and pattern-generating neurons are hypothesized to be represented by the subsets of Dbx1-positive preBötC neurons that are somatostatin-negative ($SST^-$) and -positive ($SST^+$), respectively (*Cui et al., 2016*; *Ashhad and Feldman, 2020*). Somatostatin-positive neurons are estimated to comprise 72.6% of the $Dbx1^+$ preBötC population (*Koizumi et al., 2016*). Therefore, our preBötC network was constructed such that the rhythm and pattern-forming subpopulations represent 25% and 75% of the $N = 400$ neuron preBötC population (i.e., $N_R = 100$ and $N_P = 300$). The rhythm- and pattern-generating neurons are distinguished by their $I_{NaP}$, $I_{Leak}$, and $I_{CAN}$ conductances. Also, we included the K$^+$ leak current $I_{GIRK}$ exclusively to the rhythm generating subpopulation, the activation of which we used as one representation of the effects of opioid application (*Figure 6*).

The synaptic connection probabilities within the rhythm- and pattern-generating neurons, $P_{RR} = 13\%$ and $P_{PP} = 2\%$, were taken from previous experimental findings (*Rekling et al., 2000* and *Ashhad and Feldman, 2020*, respectively). The connection probabilities between the rhythm- and pattern-generating populations are not known and in the model were set at $P_{RP} = P_{PR} = 30\%$ such that the total connection probability in the network is approximately 13% (*Rekling et al., 2000*).

Heterogeneity was introduced by normally distributing the parameters $g_{leak}$, $g_{NaP}$, and $g_{CAN}$ as well as uniformly distributing the weights ($W_{j,i}$) of excitatory synaptic connections (see *Table 2* and *Table 3*). Additionally, $g_{leak}$ was conditionally distributed with $g_{NaP}$ in order to achieve a bivariate normal distribution between these two conductances, as suggested by *Del Negro et al., 2002a* and *Koizumi and Smith, 2008*. In our simulations, this was achieved by first normally distributing $g_{NaP}$ in each neuron according to the values presented in *Table 2*. Then we used a property of bivariate

normal distribution, which says that the conditional distribution of $g_{leak}$ given $g_{NaP}$ is itself a normal distribution with mean ($\mu^*_{Leak}$) and standard deviation ($\sigma^*_{Leak}$) described as follows:

$$\mu^*_{Leak} = \mu_{Leak} + \rho \cdot (\sigma_{Leak}/\sigma_{NaP}) \cdot (g^i_{NaP} - \mu_{NaP}), \tag{27}$$

$$\sigma^*_{Leak} = \sqrt{(1 - \rho^2) \cdot \sigma^2_{Leak}} \tag{28}$$

In these equations, $\mu_{Leak}$ and $\mu_{NaP}$ are the mean and $\sigma_{Leak}$ and $\sigma_{NaP}$ are the standard deviation of the $g_{Leak}$ and $g_{NaP}$ distributions, while $\rho = 0.8$ represents the correlation coefficient and $g^i_{NaP}$ represents the persistent sodium current conductance for the $i$th neuron. All parameters are given in **Table 2**.

## Activation dynamics of $I_{Holo}$

Holographic stimulation was simulated by activating $I_{Holo}$ in small sets of randomly selected neurons across the preBötC population. Activation of this current was simulated by the following equation:

$$\frac{dm_{Holo}}{dt} = -\frac{m_{Holo}}{\tau_{Holo}} + \delta(t - t_{stim}) \tag{29}$$

where $m_{Holo}$ represents the channel activation and ranges between 0 and 1, $\tau_{Holo}$ represents the decay time constant, and $\delta(.)$ is the Kronecker delta function, which represents the instantaneous jump in $m_{Holo}$ from 0 to 1 at the time of stimulation ($t_{stim}$). Parameters were chosen such that the response in stimulated neurons matched those seen in **Kam et al., 2013b**. All parameters are given in **Table 1**.

## Data analysis and definitions

Data generated from simulations was postprocessed in MATLAB (MathWorks, Inc). An action potential was defined to have occurred in a neuron when its membrane potential $V_m$ increased through $-35\,\text{mV}$. Histograms of population activity were calculated as the number of action potentials per $20\,\text{ms}$ bin per neuron, with units of $APs/(s \cdot neuron)$. Network burst and burstlet amplitudes and frequencies were calculated by identifying the peaks and the inverse of the interpeak interval from the population histograms. The thresholds used for burst and burstlet detection were $30\,\text{spk/s/N}$ and $2.5\,\text{spk/s/N}$, respectively. For the simulated holographic stimulation simulations, the start of a network burst was defined as the time at which the integrated preBötC population activity increased through the threshold for burst detection, while the end of a network burst was defined as the time at which the integrated preBötC activity returned to exactly zero.

## Integration methods

All simulations were performed locally on an 8-core Linux-based operating system or on compute nodes at the University of Pittsburgh's Center for Research Computing. Simulation software was custom written in C++. Numerical integration was performed using the first-order Euler method with a fixed step-size ($\Delta t$) of $0.025\,\text{ms}$.

## Acknowledgements

We thank Christopher Del Negro for his comments on a draft of this manuscript.

## Additional information

### Funding

| Funder | Grant reference number | Author |
| --- | --- | --- |
| National Science Foundation | DMS1951095 | Jonathan E Rubin |

The funders had no role in study design, data collection and interpretation, or the decision to submit the work for publication.

## Author contributions
Ryan S Phillips, Conceptualization, Formal analysis, Investigation, Software, Visualization, Writing – original draft, Writing – review and editing; Jonathan E Rubin, Conceptualization, Funding acquisition, Supervision, Writing – original draft, Writing – review and editing, Formal analysis

## Author ORCIDs
Ryan S Phillips ⓘ http://orcid.org/0000-0002-8570-2348
Jonathan E Rubin ⓘ http://orcid.org/0000-0002-1513-1551

## Decision letter and Author response
Decision letter https://doi.org/10.7554/eLife.75713.sa1
Author response https://doi.org/10.7554/eLife.75713.sa2

---

## Additional files

### Supplementary files
• Transparent reporting form

### Data availability
All data generated or analysed during this study are included in the manuscript and supporting file; Source Data files have been provided for Figures 1-8. Simulation code has been published on GitHub at the following link. https://github.com/RyanSeanPhillips/Putting-the-theory-into-burstlet-theory (copy archived at swh:1:rev:fe56e63bc5839b7a6bd60f1fd0e22f8bec2a7669).

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
