## [Editor Report]

This article is of significant interest to readers in the field of neural control of breathing and for researchers interested in the generation of neuronal rhythms in general. The study assembles a sophisticated computational modeling approach to test long-standing theories and emerging views in neural control of breathing and more specifically on biophysical mechanisms of burstlet generation in the respiratory network (the preBötzinger complex network). This work is an important contribution to a better understanding of the respiratory rhythm generation, will help validate (or not) running hypotheses and will guide future experiments.

---

## [Decision Letter]

**Decision letter after peer review:**

Thank you for submitting your article "Putting the theory into 'burstlet theory': A biophysical model of bursts and burstlets in the respiratory preBötzinger complex" for consideration by *eLife*. Your article has been reviewed by 3 peer reviewers, and the evaluation has been overseen by a Reviewing Editor and Ronald Calabrese as the Senior Editor. The following individuals involved in review of your submission have agreed to reveal their identity: Nicholas Mellen (Reviewer #2); Sharmila Venugopal (Reviewer #3).

Essential revisions:

Your paper has been examined by three reviewers and myself. Below are their detailed reviews, but find also here a quick summary of their main comments that must be addressed in a revised version in order for us to judge the potential suitability of your study to be published in *eLife*.

1) Model development is based on a few groups' work, (almost entirely) based on in vitro patch-clamp techniques in neonates and slice preparations. This is extremely important to acknowledge and authors should try and add some valuable discussion on translation to in vivo situations or including a more complete part of the central respiratory command (the RTN/pFRG, pontine structures,….).

2) It is necessary for the authors to make the model more general by showing qualitatively similar CICR-mediated burstlet-to-burst amplification when alternative rhythmogenic mechanisms are used (not exclusively based on INaP for instance, considering the role of intra-network connectivity, etc…), otherwise it narrows down the model's relevance and potential application in other systems.

3) More clarity is needed in the discussion of how model predictions could guide new experiments particularly since the model sets out to challenge prevailing and proposed views of breathing rhythm and pattern generation.

4) The model code must be made available for review and upon acceptance be shared widely on GitHub and not just upload on ModelDB where many models don't even work.

*Reviewer #1 (Recommendations for the authors):*

I think for the claim to provide unifying model for respiratory rhythm generation the modelling approach of Shao J, Tsao T-H & Butera R (2006). Bursting Without Slow Kinetics: A Role for a Small World? Neural Computation 18, 2029-2035 needs to be considered. In addition work using the perfused brainstem preparation further supports the notion of network connectivity for the emergence of burstlets and burst in the phrenic motor output. See: Jones SE, Dutschmann M. Testing the hypothesis of neurodegeneracy in respiratory network function with a priori transected arterially perfused brain stem preparation of rat. J Neurophysiol. 2016 May 1;115(5):2593-607. doi: 10.1152/jn.01073.2015. Epub 2016 Feb 17. PMID: 26888109; PMCID: PMC4922475.

While I understand that it might be not possible to extend the current modelling approach to consider the network connectivity beyond its current stage it is the bare minimum to discuss the above-mentioned model and experimental data.

*Reviewer #2 (Recommendations for the authors):*

In what follows, textual revisions are suggested.

Lines 108-112: "In this computational study, we put together and build upon these previous findings to show that periodic amplification of synaptically triggered ICAN transients by calcium induced calcium release (CICR) from intracellular stores provides a plausible mechanism that can produce the observed conversion of burstlets into bursts and can explain all of the key observations underlying the burstlet theory of respiratory rhythm generation, thus providing a sound mechanistic basis for this conceptual framework.”

– This is too strong: the mechanism by which burstlets are generated at a stable frequency is stipulated using generic methods for which experimental evidence is weak.

Lines 148, 149: Increasing IAPP increases the burst frequency in neuron 1 and decreases the number of spikes per neuron 1 burst (Figure 2A3,A4), consistent with past literature (Butera et al., 1999).

– This doesn't seem to be the case for traces shown in 2C2-2C4. Also, this is same result is shown to be the case in (9).

Lines 240-244: The decrease in amplitude in the case of ICAN block is due to derecruitment of neurons from the pattern forming subpopulation and a decrease in the firing rate of the neurons that remain active, whereas in the case of ca^2+^ block the decrease in amplitude results primarily from derecruitment (Figure 5E & F). These simulations provide mechanism-specific predictions that can be experimentally tested.

– This appears to have been experimentally tested in a study cited later in your paper (lines 434-438): "In a separate study, however, block of the SERCA pump by bath application of thapsigargin (2-20 uM) or cyclopiazonic acid (CPA) (30 – 50 uM) did not significantly affect the amplitude or frequency of hypoglossal motor output in in vitro slice preparations containing the preBötC. It is possible that the negative results presented by the latter work occur due to the failure of pharmacological agents to fully penetrate the slice and diffuse across the cell membranes to reach their intracellular targets." This isn't supported by Figure 5 D, which shows that burst amplitude decreases in a graded manner with I(CAN) blockade. Thus, even if the pharmacological agents only partially penetrated the tissue, you would nonetheless expect an attenuation of amplitude.

Lines 276-278: These elicited bursts occur with delays of several hundred milliseconds relative to the stimulation time, which is longer than would be expected from existing models. Interestingly, in the current model, due to the dynamics of CICR, there is a natural delay between the onset of burstlets and the recruitment of the follower population that underlies the transition to a burst.

– For me at least, identifying model components with slower time-constants is key to developing intuitions about how models work. It might be useful to add some text to the part of the paper where you describe this component of the model (starting roughly at line 647), providing a description of the different rates of the processes that follow.

Lines 290, 291: Moreover, the probability of elicting (typo in the text) a burst increases and the delay decreases as the time after an endogenous burst increases (Figure 7G,H).

– This result seems to have little to do with CICR-related processes, and instead be due to the dynamics of your I(NaP) bursters. As such, at least some of what you report here is in fact determined by your choice of rhythmogenic mechanism, which seems to be stipulated rather than empirically grounded. This goes against the claim made in lines 605-607 quoted above.

Lines 301-303: These simulations were conducted with fixed network synaptic strength, defined as S = N(P) * P(PP)* W(PP), where W(P)P is adjusted to compensate for changes in P(PP) to keep S constant.

– What is the motivation for keeping synaptic strength fixed? Is it motivated by the biology, or for computational efficiency? Whatever the motivation, the findings you report in lines 311-319 seem to flow directly from this choice.

Lines 372-374: Our simulations support an alternative view that builds directly from previous computational studies (Jasinski et al., 2013; Phillips et al., 2019; Phillips and Rubin, 2019; Phillips et al., 2021), which robustly reproduce a wide array of experimental observations.

– The fly in the ointment here is that there was a period when endogenous bursters were presented as the mechanism for respiratory rhythmogenesis, and everyone went looking for them, and they just weren't found in sufficient numbers to carry the story. This may have been due to technical limitations, since most groups were using single-unit recording methods. Currently, the inducible dbx1-cre mouse, when crossed with a genetically encoded ca^2+^ indicator (GECI) lox mouse, will generate mice in whom most glutamatergic preBotC neurons will express the GECI. If your collaborator has access to a 2-photon electrophysiology rig (Jeff Smith has one, Chris del Negro has one), you can look for these endogenous bursters under synaptic blockade in a way that will generate robust positive or negative results, and will settle this issue (please publish regardless of outcome). In my optical recordings, under synaptic blockade, any neuron that remains active is really salient, and I'm restricted to widefield recording methods (I almost never see stationary rhythmicity in neurons that remain active), so with a 2-photon rig and a 600 μm slice, a relatively small number of experiments will give you a really robust result. Unlike the percolation model, which is probably pretty difficult to test experimentally, because it isn't really a model at all, models that assume the existence of a population of endogenous bursters are eminently testable using optical recording methods.

*Reviewer #3 (Recommendations for the authors):*

Lines 203 – 204: "In this case, the frequency of the postsynaptic ca^2+^ oscillation is again controlled by Kbath and the ca^2+^ amplitude is determined by the burstlet amplitude and PSynCa. "

Could the authors clarify which figure panel demonstrates the above statement? From the current Figures4B,C, increasing Kbath increased both frequency and amplitude. From Figure 4E-G, it is not clear what the specific contribution of PSynCa alone would be since Kbath and PSynCa are co-varied in those simulations. Also is this Figure 4 missing panel D?

Figure 6: In panel A, it seems like uOR is activated in all rhythmogenic neurons by both rhythmogenic and pattern generating neurons, whereas uOR is activated on pattern generating neurons only due to inputs from rhythmogenic neurons. This would mean that uOR is expressed throughout the preBotz abundantly in contrast with the stated references with 8-50% expression. This needs clarification. In panels B, C, what is changed from trace to trace? Is the GIRK conductance progressively increasing in B and similarly synaptic block in C? The legend does not clarify this. Could the authors also elaborate on the basis for IGIRK equation 9?

Figure 7: Can the authors comment on whether there could be a network burst if hypothetically only pattern forming neurons were stimulated?

Lines 77 – 79: "The small number of neurons required to evoke a network burst and the extended duration of the delays both differ from what would be predicted by existing computational preBötC models. "

The above is stated but not clarified why and by which models.

Lines 378 – 381: "Importantly, we find that the burstlet fraction is determined by the probability that a burstlet will trigger CICR in the pattern forming subpopulation. In the model, this probability is determined by the magnitude of postsynaptic calcium transients as well as the activation dynamics of the IP3 receptor and the SERCA pump."

Need clarification of direct/indirect evidence to support this assumption.

Lines 404 – 407: "As Kbath is increased, however, increases in the membrane potential of pattern-forming neurons and EPSP magnitude are predicted to increase the magnitude of EPSPs triggered by postsynaptic calcium transients. This is exactly the effect that is captured in the model by an increase in PSynCa."

The above statements need refinement. The parameter PsynCa is not linked to Kbath in the model. Therefore, the proposition that PsynCa 'captures' two distinct processes: (1) Increase in RMP caused by increased Kbath, (2) secondary to that, an increase in EPSP magnitude does not seem right. If they are linked to K_bath in the model, then it is justified that the model indeed tested the link between K_bath changes and EPSP amplitudes. Currently this is not the case.

Lines 431 – 438: "For example, Mironov (2008) showed that the transmission of calcium waves that travel from the dendrites to the soma is blocked by local application of thapsigargin, a SERCA pump inhibitor. In a separate study, however, block of the SERCA pump by bath application of thapsigargin (2 – 20 uM) or cyclopiazonic acid (CPA) (30 – 50 uM) did not significantly affect the amplitude or frequency of hypoglossal motor output in in vitro slice preparations containing the preBötC. It is possible that the negative results presented by the latter work occur due to the failure of pharmacological agents to fully penetrate the slice and diffuse across the cell membranes to reach their intracellular targets."

Here, the authors speculate that lack of an effect of THIP or CPA is due to lack of sufficient drug penetration and no reference has been cited. Without an appropriate experimental validation using suitable concentrations, drug perfusion durations and slice thickness, such a proposal seems conjectural to suggest in a purely modeling study. Lack of an effect could also be due to low sample size, small effect size among other possible differences in experimental conditions.

Lines 462 – 470: Not entirely certain what the authors are trying to compare here with exact values of Kbath across model and experiments. This seems irrelevant as the model does not have all the biological attributes and experimental conditions. It would instead be reasonable to discuss whether there were qualitative differences in the model behavior compared to experiments in the range of Kbath values which represent concentrations similar to those used in experiments.

---

## [Author Response]

Essential revisions:Your paper has been examined by three reviewers and myself. Below are their detailed reviews, but find also here a quick summary of their main comments that must be addressed in a revised version in order for us to judge the potential suitability of your study to be published in eLife.1) Model development is based on a few groups' work, (almost entirely) based on in vitro patch-clamp techniques in neonates and slice preparations. This is extremely important to acknowledge and authors should try and add some valuable discussion on translation to in vivo situations or including a more complete part of the central respiratory command (the RTN/pFRG, pontine structures,….).

We discuss corresponding revisions below in our response to Reviewer #1. Text has been added to the Discussion, especially on pg. 19, to address this issue.

2) It is necessary for the authors to make the model more general by showing qualitatively similar CICR-mediated burstlet-to-burst amplification when alternative rhythmogenic mechanisms are used (not exclusively based on INaP for instance, considering the role of intra-network connectivity, etc…), otherwise it narrows down the model's relevance and potential application in other systems.

We generated new simulations showing qualitatively similar CICR-mediated burst-to-burstlet recruitment with an imposed rhythm within the rhythmogenic subpopulation; see the new Figure 4—figure supplement 1, which is now referenced on Lines 222-224 and Lines 660-662. We discuss the corresponding revisions below in response to Reviewer #2.

3) More clarity is needed in the discussion of how model predictions could guide new experiments particularly since the model sets out to challenge prevailing and proposed views of breathing rhythm and pattern generation.

Our work presents a model. At a high level, the model itself is the prediction; that is, the work predicts that a CICR-mediated mechanism is critical to the translation of burstlets into bursts. At a more specific level, our manuscript makes the following predictions:

– The magnitude of postsynaptic calcium transients triggered by EPSPs in preBotC inspiratory neurons will increase with Kbath; see Figure 4, related text, and lines 419-421 of the discussion.

– Network-level burstlets and bursts will persist under block of Ih and IA; see lines 516-521.

– Blocking calcium transients will increase the burstlet fraction and decrease burst amplitudes; see Figure 5 and related text.

– ICAN block will not change the burstlet fraction and will decrease burst amplitudes; see Figure 5 and related text.

– The synaptic connection probability within the pattern-generating population in the preBotC is low (1-2%); see Figure 8 and related text.

– Selective holographic stimulation of pattern forming neurons should be more effective at triggering network bursts; see Figure 8 and related text. This could be tested by selectively stimulating Dbx1 preBotC neurons that express Sst (pattern forming) or that do not express Sst (rhythmogenic).

We have added a paragraph to the Discussion to recap these predictions so that they are more apparent to readers; see lines 665-678.

4) The model code must be made available for review and upon acceptance be shared widely on GitHub and not just upload on ModelDB where many models don't even work.

Model code is now available on GitHub at the following link:

https://github.com/RyanSeanPhillips/Putting-the-theory-into-burstlet-theory

Reviewer #1 (Recommendations for the authors):I think for the claim to provide unifying model for respiratory rhythm generation the modelling approach of Shao J, Tsao T-H & Butera R (2006). Bursting Without Slow Kinetics: A Role for a Small World? Neural Computation 18, 2029-2035 needs to be considered. In addition, work using the perfused brainstem preparation further supports the notion of network connectivity for the emergence of burstlets and burst in the phrenic motor output. See: Jones SE, Dutschmann M. Testing the hypothesis of neurodegeneracy in respiratory network function with a priori transected arterially perfused brain stem preparation of rat. J Neurophysiol. 2016 May 1;115(5):2593-607. doi: 10.1152/jn.01073.2015. Epub 2016 Feb 17. PMID: 26888109; PMCID: PMC4922475.While I understand that it might be not possible to extend the current modelling approach to consider the network connectivity beyond its current stage it is the bare minimum to discuss the above mentioned model and experimental data.

In fact, the paper by Shao et al. uses the Morris-Lecar model as its representation of each individual neuron in the network. The network consists of a linear chain of neurons and the neurons at the end are oscillators, which is the state of the Morris-Lecar model that corresponds to endogenous bursting. Those oscillators/pacemakers are responsible for initiating the activity in the network on each cycle. The architectural results in that work are not highly relevant for the respiratory network. That is, these results show that if neurons are aligned in a linear chain with the pacemakers on the ends, then activity propagates through the network in waves, whereas adding enough random, long-range connections (so that the network qualifies as “small world”) allows the activity to arise more uniformly throughout the network. There is no evidence to suggest anything like this arrangement of connections in the preBotC respiratory circuit.

We do, however, acknowledge that there may be an important role for the network architecture and for inputs from elsewhere in the respiratory brain stem (possibly as sources of drive that sets excitability) in respiratory rhythm generation and, as shown by Jones and Dutschmann, in pattern generation. Thus, we have edited our Discussion on pg. 19 to make these points clear and to provide some relevant references to the literature.

As a final point, we also recognize that our model is not truly unifying, in that it does not pull together every idea that has been put forward as a possible factor in respiratory rhythm and pattern generation, and hence we have changed the wording in our Introduction and Discussion to avoid a claim of full unification. For example, our Introduction now ends with the statement “In this computational study, we put together and build upon these previous findings to show that periodic amplification of synaptically triggered ICAN transients by calcium induced calcium release (CICR) from intracellular stores provides a plausible mechanism that can produce the observed conversion of burstlets into bursts and can explain diverse experimental findings associated with this process. Altogether, our modeling work suggests a plausible mechanistic basis for the conceptual framework of burstlet theory and the experimental observations that this theory seeks to address.” We hope that this less expansive statement provides a more accurate representation of what our work achieves.

Reviewer #2 (Recommendations for the authors):In what follows, textual revisions are suggested.Lines 108-112: "In this computational study, we put together and build upon these previous findings to show that periodic amplification of synaptically triggered ICAN transients by calcium induced calcium release (CICR) from intracellular stores provides a plausible mechanism that can produce the observed conversion of burstlets into bursts and can explain all of the key observations underlying the burstlet theory of respiratory rhythm generation, thus providing a sound mechanistic basis for this conceptual framework.– This is too strong: the mechanism by which burstlets are generated at a stable frequency is stipulated using generic methods for which experimental evidence is weak.

We apologize for the overreach in our wording. We have revised this text (now lines 115-117), as well as the abstract, where similar statements were made in the submitted manuscript, to weaken our claims and to more clearly emphasize the CICR aspect of the model.

Lines 148, 149: Increasing IAPP increases the burst frequency in neuron 1 and decreases the number of spikes per neuron 1 burst (Figure 2A3,A4), consistent with past literature (Butera et al., 1999).– This doesn't seem to be the case for traces shown in 2C2-2C4. Also, this is same result is shown to be the case in (9).

In progressing from Figure 2C2-2C4, both Iapp and Psynca are varied. The effects of varying Iapp alone, including the decrease in number of spikes per burst with increasing Iapp, are shown in Figure 2A3, 2A4, which we do reference in this sentence (now lines 154-156). We have added a reference to (9) in the corresponding text to go along with Butera et al., 1999.

Lines 240-244: The decrease in amplitude in the case of ICAN block is due to derecruitment of neurons from the pattern forming subpopulation and a decrease in the firing rate of the neurons that remain active, whereas in the case of ca^2+^ block the decrease in amplitude results primarily from derecruitment (Figure 5E & F). These simulations provide mechanism-specific predictions that can be experimentally tested.– This appears to have been experimentally tested in a study cited later in your paper (lines 434-438): "In a separate study, however, block of the SERCA pump by bath application of thapsigargin (2-20 uM) or cyclopiazonic acid (CPA) (30 – 50 uM) did not significantly affect the amplitude or frequency of hypoglossal motor output in in vitro slice preparations containing the preBötC. It is possible that the negative results presented by the latter work occur due to the failure of pharmacological agents to fully penetrate the slice and diffuse across the cell membranes to reach their intracellular targets." This isn't supported by Figure 5 D, which shows that burst amplitude decreases in a graded manner with I(CAN) blockade. Thus, even if the pharmacological agents only partially penetrated the tissue, you would nonetheless expect an attenuation of amplitude.

The effects of ICAN block on hypoglossal output in in vitro slice preparations (i.e. preBotC bursts) have been experimentally tested (Koizumi et al., eNeuro 2018 and Picardo et al., PLoS Biology 2019). Both studies found that ICAN block decreased amplitude and only slightly decreased or had no impact on frequency, as is the case if Figure 5B & D with ICAN block. These points are now highlighted on lines 450-454 of the Discussion.

Application of thapsigargin or cyclopiazonic acid (discussed on lines 434-438 of the original manuscript) blocks the SERCA pump, which is involved in ca^2+^ uptake from the cytoplasm into the ER. This is not equivalent to ICAN block or the block of ca^2+^ transients simulated in Figure 5. Simulating the effects of SERCA pump block may be complicated, as this could lead to calcium build up in the cytoplasm and (in)activation of multiple cellular mechanisms. However, if the SERCA pump was nonuniformly blocked across the network due to incomplete drug penetration we would expect some impact on rhythmicity as suggested by this reviewer. Therefore, we have removed the line of text about incomplete slice penetration from the Discussion. Importantly, application of 1uM thapsigargin in Mironov et al., *Journal of Physiology*, 2008 does stop rhythmic activity, which is now noted on lines 467-469, with additional relevant edits in lines 473-475.

Lines 276-278: These elicited bursts occur with delays of several hundred milliseconds relative to the stimulation time, which is longer than would be expected from existing models. Interestingly, in the current model, due to the dynamics of CICR, there is a natural delay between the onset of burstlets and the recruitment of the follower population that underlies the transition to a burst.– For me at least, identifying model components with slower time-constants is key to developing intuitions about how models work. It might be useful to add some text to the part of the paper where you describe this component of the model (starting roughly at line 647), providing a description of the different rates of the processes that follow.

We have added new text in the Methods section, on lines 449-750 and 770-771, to indicate the relative timescales of the processes in the model and to provide relevant references with more details.

Lines 290, 291: Moreover, the probability of elicting (typo in the text) a burst increases and the delay decreases as the time after an endogenous burst increases (Figure 7G,H).– This result seems to have little to do with CICR-related processes, and instead be due to the dynamics of your I(NaP) bursters. As such, at least some of what you report here is in fact determined by your choice of rhythmogenic mechanism, which seems to be stipulated rather than empirically grounded. This goes against the claim made in lines 605-607 quoted above.

Multiple factors in our model contribute to this delay, including network connectivity, calcium dynamics, and persistent sodium current dynamics. We have added new lines 313-320 in the Results section, along with a new supplemental figure (Figure 7 – Supp. Figure 1), to indicate which processes within the model contribute to which aspects of the stimulation effects. We have also fixed the typo in these lines.

Lines 301-303: These simulations were conducted with fixed network synaptic strength, defined as S = N(P) * P(PP)* W(PP), where W(P)P is adjusted to compensate for changes in P(PP) to keep S constant.– What is the motivation for keeping synaptic strength fixed? Is it motivated by the biology, or for computational efficiency? Whatever the motivation, the findings you report in lines 311-319 seem to flow directly from this choice.

Sorry for not being clear about this choice. This was a deliberate computational experiment, where fixing the synaptic strength provided us a way to examine the effects of changes in connection probability in a fair way, without also changing the overall input strengths to neurons. The reviewer is correct that the findings in lines 311-319 flow from this choice, and that is by design. We have slightly revised the wording in lines 327-329 to clarify the reason for the manner in which we implemented this study.

Lines 372-374: Our simulations support an alternative view that builds directly from previous computational studies (Jasinski et al., 2013; Phillips et al., 2019; Phillips and Rubin, 2019; Phillips et al., 2021), which robustly reproduce a wide array of experimental observations.– The fly in the ointment here is that there was a period when endogenous bursters were presented as the mechanism for respiratory rhythmogenesis, and everyone went looking for them, and they just weren't found in sufficient numbers to carry the story. This may have been due to technical limitations, since most groups were using single-unit recording methods. Currently, the inducible dbx1-cre mouse, when crossed with a genetically encoded ca^2+^ indicator (GECI) lox mouse, will generate mice in whom most glutamatergic preBotC neurons will express the GECI. If your collaborator has access to a 2-photon electrophysiology rig (Jeff Smith has one, Chris del Negro has one), you can look for these endogenous bursters under synaptic blockade in a way that will generate robust positive or negative results, and will settle this issue (please publish regardless of outcome). In my optical recordings, under synaptic blockade, any neuron that remains active is really salient, and I'm restricted to widefield recording methods (I almost never see stationary rhythmicity in neurons that remain active), so with a 2-photon rig and a 600 μm slice, a relatively small number of experiments will give you a really robust result. Unlike the percolation model, which is probably pretty difficult to test experimentally, because it isn't really a model at all, models that assume the existence of a population of endogenous bursters are eminently testable using optical recording methods.

We will certainly discuss these tests with our experimental collaborators. Hopefully our work will help inspire future testing of these ideas.

Reviewer #3 (Recommendations for the authors):Lines 203 – 204: "In this case, the frequency of the postsynaptic ca^2+^ oscillation is again controlled by Kbath and the ca^2+^ amplitude is determined by the burstlet amplitude and PSynCa. "Could the authors clarify which figure panel demonstrates the above statement? From the current Figures4B,C, increasing Kbath increased both frequency and amplitude. From Figure 4E-G, it is not clear what the specific contribution of PSynCa alone would be since Kbath and PSynCa are co-varied in those simulations. Also is this Figure 4 missing panel D?

Figure 4 panel labels have been revised and references to Figure 4 panels have been updated in the text.

Lines 203-204 of the original manuscript were not clear as to which figure panels justify the claims in this statement. This text, now on lines 215-221, has been updated to read: "In this case, the frequency of the postsynaptic ca^2+^ oscillation is controlled by Kbath (Figure 4B). However, because Kbath also affects burstlet amplitude (Figure 4C), the postsynaptic ca^2+^ amplitude is determined by both Kbath and PSynCa. If Kbath is held fixed, then modulating Psynca will only affect the amplitude of the postsynaptic ca^2+^ transient, since burstlet amplitude will not be impacted. The effects of selectively changing the postsynaptic ca^2+^ amplitude on the full network can thus be extracted by considering a vertical slice through Figure 4 E-F."

Figure 6: In panel A, it seems like uOR is activated in all rhythmogenic neurons by both rhythmogenic and pattern generating neurons, whereas uOR is activated on pattern generating neurons only due to inputs from rhythmogenic neurons. This would mean that uOR is expressed throughout the preBotz abundantly in contrast with the stated references with 8-50% expression. This needs clarification. In panels B, C, what is changed from trace to trace? Is the GIRK conductance progressively increasing in B and similarly synaptic block in C? The legend does not clarify this. Could the authors also elaborate on the basis for IGIRK equation 9?

The uOR label on the rhythm generating population has been moved to the left side to avoid giving readers the faulty impression that the synapses from the pattern forming population to the rhythm generating population are expressing uOR. Some text (in red) has been added to the figure caption to indicate that the manipulations in panels B and C are increasing from top to bottom traces. Finally, we have revised the text on line 272, line 701 and lines 790-792 to clarify that IGIRK is only included in the rhythmogenic neural subpopulation in the model and is a form of potassium leak current, hence the form of equation (9).

Figure 7: Can the authors comment on whether there could be a network burst if hypothetically only pattern forming neurons were stimulated?

Yes, a network burst can be triggered when only pattern forming neurons are stimulated. This is shown in Figure 8H. In fact, the probability of triggering burst is highest if a greater proportion of pattern forming neurons are stimulated. Moreover, pattern forming neurons are more likely to be stimulated since they out number rhythmogenic neurons. These points are now discussed on lines 368-371.

Lines 77 – 79: "The small number of neurons required to evoke a network burst and the extended duration of the delays both differ from what would be predicted by existing computational preBötC models. "The above is stated but not clarified why and by which models.

We agree with the reviewer that we were overly cavalier in making this statement. We have rewritten these lines of text (now lines 78-82) to be more precise and to include citations to relevant literature including modeling papers.

Lines 378 – 381: "Importantly, we find that the burstlet fraction is determined by the probability that a burstlet will trigger CICR in the pattern forming subpopulation. In the model, this probability is determined by the magnitude of postsynaptic calcium transients as well as the activation dynamics of the IP3 receptor and the SERCA pump."Need clarification of direct/indirect evidence to support this assumption.

We apologize for the confusion caused by our wording. Here we are explaining/summarizing the mechanisms in the model that determine the burstlet fraction via CICR. More specifically, we make the point that the dynamics of CICR are controlled by the calcium dependent activation dynamics of the IP3 receptor and the SERCA pump, which regulate the rate of ca^2+^ uptake and release from ER stores.

We have edited the text on line 411 to clarify that this control of the burstlet fraction is a model prediction.

Lines 404 – 407: "As Kbath is increased, however, increases in the membrane potential of pattern-forming neurons and EPSP magnitude are predicted to increase the magnitude of EPSPs triggered by postsynaptic calcium transients. This is exactly the effect that is captured in the model by an increase in PSynCa."The above statements need refinement. The parameter PsynCa is not linked to Kbath in the model. Therefore, the proposition that PsynCa 'captures' two distinct processes: (1) Increase in RMP caused by increased Kbath, (2) secondary to that, an increase in EPSP magnitude does not seem right. If they are linked to K_bath in the model, then it is justified that the model indeed tested the link between K_bath changes and EPSP amplitudes. Currently this is not the case.

This statement has been revised to improve clarity. The revised text, now in lines 438-442, reads “As Kbath is increased, however, both EPSC magnitudes and the membrane potential of pattern-forming neurons increase. Therefore, with increased Kbath, the prediction is that EPSCs will result in greater activation of voltage-gated ca^2+^ channels and increased postsynaptic calcium transients. This effect is captured in the model by an increase in the parameter PSynCa, which determines the percentage of the postsynaptic current carried by ca^2+^ ions, with Kbath."

Lines 431 – 438: "For example, Mironov (2008) showed that the transmission of calcium waves that travel from the dendrites to the soma is blocked by local application of thapsigargin, a SERCA pump inhibitor. In a separate study, however, block of the SERCA pump by bath application of thapsigargin (2 – 20 uM) or cyclopiazonic acid (CPA) (30 – 50 uM) did not significantly affect the amplitude or frequency of hypoglossal motor output in in vitro slice preparations containing the preBötC. It is possible that the negative results presented by the latter work occur due to the failure of pharmacological agents to fully penetrate the slice and diffuse across the cell membranes to reach their intracellular targets."Here, the authors speculate that lack of an effect of THIP or CPA is due to lack of sufficient drug penetration and no reference has been cited. Without an appropriate experimental validation using suitable concentrations, drug perfusion durations and slice thickness, such a proposal seems conjectural to suggest in a purely modeling study. Lack of an effect could also be due to low sample size, small effect size among other possible differences in experimental conditions.

Please refer to our comments to Reviewer #2, who raised a very similar point. In brief, we have removed the line about slice penetration. We have also added and edited relevant text in the paragraph starting on line 467.

Lines 462 – 470: Not entirely certain what the authors are trying to compare here with exact values of Kbath across model and experiments. This seems irrelevant as the model does not have all the biological attributes and experimental conditions. It would instead be reasonable to discuss whether there were qualitative differences in the model behavior compared to experiments in the range of Kbath values which represent concentrations similar to those used in experiments.

We agree with the reviewer that it is not realistic to try to match the exact values of Kbath between our model and existing experiments. However, we also recognize that this comparison may be important to experimentalists who work on respiratory neurodynamics; indeed, Reviewer 2 asked specifically about this comparison. Thus, we feel that it is important to keep this text in the manuscript.

To clarify: This text (now on lines 499-511) discusses two differences between experimental data and our simulation that can be explained by a biological feature that has been experimentally documented but was not included in the model. The first difference is that the Kbath range at which burstlets first emerge in our model is at approximately 5mM whereas in experiments burstlets persist even at 3mM. The second difference between experiments and our simulations is that in experiments, the burstlet fraction is constant when Kbath is varied between 3mM and 5mM whereas our model cannot generate oscillatory behavior below 5mM. These two differences can be explained by considering the extracellular potassium buffering capacity of respiratory circuits. Okada et al., 2005 showed that respiratory circuits appear to have some extracellular potassium buffering capacity such that for Kbath concentrations below approximately 5mM the extracellular potassium concentration remains elevated above Kbath. This suggests that, in experiments, when Kbath is 3mM the extracellular potassium concentration that neurons encounter within the slice remains elevated above Kbath. If extracellular potassium remains elevated and relatively constant with changes in Kbath between 3mM and 5mM, this would explain why burstlets persist at 3mM and why the burstlet fraction does not change over this Kbath range. Stated another way, if extracellular potassium buffering was included in the model, then this would extend the bath range at which burstlets occur and would result in a relatively constant burstlet fraction over this lower Kbath range as is seen experimentally.

As Figure 4I shows, our model does compare favorably with experimental results over the range of Kbath values relevant to both. Moreover, Figure 4H also at least qualitatively matches experimental data. We have briefly revised this part of the Discussion to also include mention of this agreement (lines 509-511).